



# Estimation of CO$_2$ fluxes in the cities of Zurich and Paris using the ICON-ART CTDAS inverse modelling framework

Nikolai Ponomarev[1], Michael Steiner[1,2], Erik Koene[1], Pascal Rubli[1], Stuart Grange[1,3], Lionel Constantin[1], Michel Ramonet[4], Leslie David[5], Lukas Emmenegger[1], and Dominik Brunner[1]

[1]Empa, Swiss Federal Laboratories for Materials Science and Technology, Laboratory for Air Pollution/Environmental Technology, Überlandstrasse 129, 8600 Dübendorf, Switzerland
[2]now at: Environmental Defense Fund, Amsterdam, The Netherlands
[3]now at: University of Bern, Physics Institute, Climate and Environmental Physics, Sidlerstrasse 5, 3012 Bern, Switzerland
[4]Laboratoire des Sciences du Climat et de l'Environnement (LSCE-IPSL), CEA-CNRS-UVSQ, Université Paris-Saclay, F-91191 Gif-sur-Yvette, France
[5]Airparif, Association Agréée pour la Surveillance de la Qualité de l'Air en région Île-de-France, 7 rue Crillon, 75004 Paris, France

**Correspondence:** Nikolai Ponomarev (nikolai.ponomarev@empa.ch) and Dominik Brunner (dominik.brunner@empa.ch)

**Abstract.** Observation-based estimation of urban CO$_2$ emissions can help cities track their pathway to net zero emissions, a goal many cities worldwide have adopted. While mesoscale atmospheric transport models are an effective component in inversion systems estimating country-level emissions, their use in urban-scale inversions presents a significant challenge. Here, we present one-year flux inversion results with the mesoscale ICON-ART atmospheric transport model for two cities with

5    contrasting size and topographic complexity: Zurich and Paris. Inversions were performed with an ensemble square root filter, assimilating observations from a dense rooftop CO$_2$ sensor network in Zurich and from a tall tower network in Paris. The inversion framework optimized gridded anthropogenic and biospheric fluxes, along with background mole fractions from eight inflow regions. Prior anthropogenic emissions were based on detailed inventories provided by local authorities. In Zurich, the inversion resulted in a posterior annual anthropogenic emission of 1012.3 $\pm$ 38.8 kt yr$^{-1}$, representing approximately a 30%

10    reduction compared to the prior, with the most significant decreases during winter periods of elevated ambient temperatures. In contrast, the posterior fluxes in Paris remained close to the prior, with an annual emission of 3580.0 $\pm$ 101.9 kt yr$^{-1}$, which is 7% higher than the prior. This comparison highlights the influence of city-specific factors—such as topography, city size, and observational network—on the inversion system performance. Furthermore, our findings demonstrate the potential of mesoscale models to refine urban emission estimates, offering valuable insights for policymakers and researchers working to

15    improve emission inventories and advance urban climate strategies.

## 1 Introduction

Inversion of CO$_2$ fluxes using atmospheric transport models is a well-established approach that was originally applied at global scale to constrain the global carbon budget (Gurney et al., 2002; Davis et al., 2021; Broquet et al., 2013; Monteil et al., 2020; van der Woude et al., 2023). With the development of regional measurement networks and advances in high-resolution



modeling, this approach has since been extended to continental and national scales. However, despite growing interest, only a limited number of inverse modeling studies have been performed at the urban scale (Lauvaux et al., 2016; Kunik et al., 2019; Sargent et al., 2018; Nalini et al., 2022). Urban inversions are gaining increasing attention, as cities are major contributors to anthropogenic emissions, making accurate emission estimates at this scale essential for supporting climate action plans and verifying reported emission reductions.

In this study, we present results from year-long $CO_2$ flux inversions conducted for two European cities, Zurich and Paris. This work is part of the European ICOS Cities project, which aimed to develop and evaluate different $CO_2$ emission monitoring systems in three pilot cities of contrasting size and topographic complexity - Paris, Munich and Zurich. The primary objective of our study is to generate robust estimates of anthropogenic $CO_2$ emissions in Zurich and Paris, using high-resolution atmospheric transport simulations in combination with dense urban $CO_2$ observation networks.

Previous studies estimating urban $CO_2$ emissions have highlighted several key challenges. One major difficulty is to differentiate between $CO_2$ enhancements caused by local sources and those resulting from inflow from surrounding regions (Lauvaux et al., 2016; Sargent et al., 2018; Broquet et al., 2013). This is because the enhancements in $CO_2$ mole fractions due to biospheric and anthropogenic sources within the domain are often of similar amplitude as the variations in the background levels. As a result, it is crucial to include observation sites located outside - and ideally upwind - of the urban area to better constrain background conditions. In this study, we address this issue by jointly optimizing background mole fractions along with anthropogenic and biospheric fluxes. Without this simultaneous optimization, any bias in the background would propagate directly into the estimated fluxes, compromising the accuracy of the inversion.

Another challenge is the separation of anthropogenic and biospheric contributions (Lauvaux et al., 2016; Sargent et al., 2018; Kunik et al., 2019; Wu et al., 2018), as their atmospheric signals often overlap in both space and time. To disentangle these sources, we rely on their distinct diurnal, seasonal, and spatial patterns. Nevertheless, this remains a major source of uncertainty—particularly during summer months when biospheric activity peaks.

A final challenge relates to the prior information on fluxes and their uncertainties (Broquet et al., 2013; Lauvaux et al., 2016; Sargent et al., 2018; Kunik et al., 2019). In our case, we could benefit from very detailed city inventories provided by the authorities of Zurich and Paris. However, temporal profiles were not always available, necessitating the use of generic time profiles in some cases. To account for these limitations and avoid introducing biases into the optimization, relatively large uncertainties were prescribed for all anthropogenic and biospheric sources, allowing the system to adjust the fluxes based on observations. At the same time, spatial correlations were introduced in the prior error covariance matrices to limit the degrees of freedom and prevent overfitting.

The measurement networks used in the inversions differed significantly between Zurich and Paris, reflecting their contrasting geographic and urban characteristics. In Paris, located in the flat Île-de-France region, a tower-based network of 9 sites with high-precision instruments was operated. The network was designed to measure both upwind background mole fractions and downwind increments due to urban emissions. Depending on prevailing wind direction, several sites alternated between upwind and downwind roles.





In contrast, Zurich's complex topography—characterized by intersecting valleys and surrounding ridges—makes it difficult
to consistently define upwind and downwind locations. To address this, a denser network of 13 mid-cost rooftop sensors was
installed within the city to monitor urban $CO_2$ enhancements, complemented by three high-precision instruments mounted on
towers outside the urban area to provide background measurements.

For the inversion, we used the mesoscale atmospheric transport model ICON-ART (Zängl et al., 2015; Rieger et al., 2015;
Schröter et al., 2018) coupled with the CarbonTracker Data Assimilation Shell (CTDAS) (Peters et al., 2005; van der Laan-
Luijkx et al., 2017). This coupling of ICON-ART and CTDAS, originally developed for methane inversions at the European
scale (Steiner et al., 2024b), was extended in this study to run $CO_2$ inversions over urban domains and to modify biospheric
fluxes "online" (i.e., during the simulation, Jähn et al., 2020). The ICON-ART simulations were performed at a spatial resolu-
tion fine enough to resolve the main topographic features of the Zurich area, including the Limmattal and Glattal valleys and
the surrounding ridges, which rise 100–400 meters above the valley floors.

In this study, we adress the aforementioned challenges by applying a high-resolution inversion framework that jointly opti-
mizes anthropogenic emissions, biospheric $CO_2$ fluxes, and background concentrations for two cities with markedly different
characteristics. The comparison spans a relatively straightforward case - Paris, a large, isolated city in flat terrain - and a more
complex scenario - Zurich, a mid-sized city embedded in mountainous terrain and surrounded by other urban agglomerations.
By comparing results from Zurich and Paris, we explore how sensor network design, atmospheric transport, and prior flux
uncertainties influence inversion performance.

## 2 Data and Methods

### 2.1 Observations

Atmospheric $CO_2$ dry air mole fraction measurements for both cities were obtained from ground-based station networks (see
Fig. 1). For brevity, we will refer to dry air mole fractions simply as mole fractions. The measurement data were first aggregated
to hourly values and then averaged over afternoon hours (11:00–16:00 UTC, i.e., 12:00–17:00 local time) before being used
for model evaluation and flux inversions. It is common practice to consider only daytime or afternoon measurements when
comparing simulated and observed mole fractions, as stable nocturnal boundary layers are challenging to simulate and often
lead to large model-observation mismatches (Gerbig et al., 2003).

In Zurich, the observation network comprised 13 rooftop measurement sites inside the city equipped with mid-cost sensors.
Here we use the term mid-cost sensors to distinguish them from high-precision analyzers (e.g., cavity ring-down spectroscopy)
and from low-cost sensors such as those deployed in the Carbosense network (Müller et al., 2020). As described in (Grange
et al., 2025), three different models of non-dispersive infrared (NDIR) sensors were deployed in Zurich's mid-cost sensor
network called ZiCOS-M. Different from the sensors in Zurich's low-cost sensor network ZiCOS-L, which was operated in
parallel (Creman et al., 2025), the mid-cost sensors had higher sensitivity, were mostly operated in temperature-controlled
rooms, and were calibrated daily by supplying calibration gas from two reference gas cylinders. Their accuracy was about 1
ppm (Grange et al., 2025), which is one order of magnitude better than the accuracy of the low-cost units. Three background





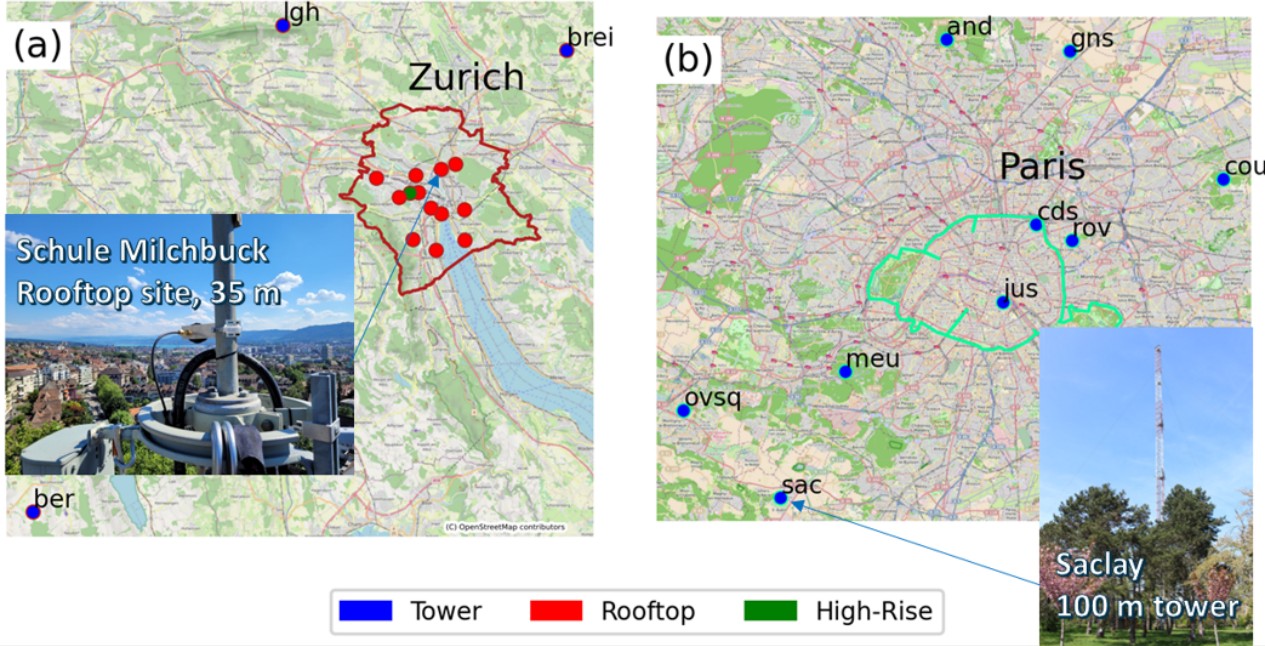

**Figure 1.** Locations of the measurement stations comprising the Zurich (a) and Paris (b) $CO_2$ observation networks. Tower sites in both cities are labeled with their acronyms. Photo of the Zurich Schule Milchbuck station by Pascal Rubli; photo of the Paris Saclay station by Michel Ramonet. Basemap tiles for panels (a) and (b) © OpenStreetMap contributors 2025. Distributed under the Open Data Commons Open Database License (ODbL) v1.0.

sites were located outside the city. Two of the background sites were equipped with high-precision instruments, one with a mid-cost sensor. An overview of the sites is provided in Table 1. Hardau II (hard) is a central site with a 20 m long mast mounted on top of a 95.3 m building. The sampling at this site occurs at a much higher altitude compared to the other rooftops in the network (Grange et al., 2025). This site, which features additional monitoring activities including an eddy covariance system, is highlighted in Fig.1 as high-rise to indicate its elevated inlet height. The mid-cost sensor network consisted of 19 sites inside the city, but six of these were discarded because they either measured at street-level or were influenced by local sources. Both situations cannot be captured adequately by a mesoscale atmospheric transport model. Data from the stations were subject to basic quality control before being used in the inversion. A full description of the Zurich sensor network, calibration strategy and data processing is provided in Grange et al. (2025). The inversion period spanned a full year from September 2022 to August 2023, preceded by a two-week spin-up in August 2022.

In Paris, all measurements used in this work were recorded at nine tower sites equipped with high-precision instruments (Doc et al., 2024). Most of the sites were located outside the city, with three measuring upwind mole fractions and three measuring downwind ones, while the remaining three were located within the city. The sites are listed in Tab. 2. The inversion for Paris also covered a full year, but for the period between January 2023 to December 2023. A detailed description of the measurement network and data handling procedures is available in (Doc et al., 2024). A mid-cost $CO_2$ sensor network was





**Table 1.** ICOS Cities sites in Zurich with mid-cost sensors installed on rooftops and 2 background sites (Beromünster and Laegern-Hochwacht) equipped with high-precision instruments. Elevation refers to height above sea level, and inlet height to height above ground level.

| Acronym | Name | Longitude | Latitude | Elevation (m) | Inlet height (m) | Instrument |
|---------|------|-----------|----------|---------------|------------------|------------|
| zhab | Albisgüetli | 8.5128 | 47.3535 | 469.8 | 22.1 | mid-cost |
| zbas | Badenerstrasse Farbhof | 8.4803 | 47.3904 | 399.6 | 22.5 | mid-cost |
| zubv | Bankenviertel Bleicherweg | 8.5380 | 47.3689 | 408.7 | 26.5 | mid-cost |
| ber | Beromuenster | 8.1755 | 47.1896 | 797.0 | 212.0 | high-precision |
| brei | Birchwil Turm | 8.6492 | 47.4672 | 592.2 | 54.0 | mid-cost |
| zgub | Güterbahnhof | 8.5176 | 47.3817 | 407.5 | 29.4 | mid-cost |
| hard | Hardau II | 8.5102 | 47.3813 | 409.4 | 110.3 | mid-cost |
| zhhm | Hardturmstrasse Förrlibuck | 8.5153 | 47.3920 | 401.2 | 40.6 | mid-cost |
| zhhf | Kantonales Labor Zürich | 8.5585 | 47.3713 | 451.8 | 20.4 | mid-cost |
| lgh | Laegern-Hochwacht | 8.3973 | 47.4822 | 840.0 | 32.0 | high-precision |
| ztle | Letzigraben Telefonzentrale | 8.5005 | 47.3788 | 411.8 | 24.0 | mid-cost |
| zhmi | Schule Milchbuck | 8.5378 | 47.3957 | 477.7 | 35.3 | mid-cost |
| zhsf | Stauffacherstrasse Werdplatz | 8.5289 | 47.3724 | 411.4 | 48.0 | mid-cost |
| ztie | Tiefenbrunnen Wildbachstrasse | 8.5589 | 47.3530 | 408.7 | 38.8 | mid-cost |
| zhui | Universität Zürich Irchel | 8.5506 | 47.3987 | 491.7 | 29.0 | mid-cost |
| zhwh | Wollishofen | 8.5333 | 47.3470 | 407.9 | 40.6 | mid-cost |

recently also established in Paris (Lian et al., 2024). These sites were excluded from this study due to upgrades and expansion under the ICOS Cities project, but they will be available for future studies.

The dense rooftop sensor network in Zurich offers detailed spatial coverage and high sensitivity to emission sources within the city. In contrast, the tall tower network in Paris is only sensitive to emissions integrated over larger portions of the city.

## 2.2 ICON-ART model

ICON-ART (ICOsahedral Nonhydrostatic model with Aerosols and Reactive Trace gases) is a mesoscale meteorology and atmospheric transport model that consists of two components: the climate and weather prediction model ICON (Zängl et al., 2015), developed by the German Weather Service (DWD) and the Max Planck Institute for Meteorology, and ART (Rieger et al., 2015; Schröter et al., 2018; Hoshyaripour et al., 2025), mainly created by the Karlsruhe Institute of Technology, for simulations of passive and chemically reactive tracers. ICON is a versatile model that can be run from global to regional and even to sub-kilometer scale. It operates on a semi-structured grid with triangular grid cells and offers options for online (through regional grid refinement) and offline nesting. Here, we use ICON in limited-area configurations with offline nesting. ICON-ART is a fully coupled model jointly simulating weather and atmospheric tracer transport in a consistent way (Baklanov





**Table 2.** Sites from the tower network in Paris. All sites are equipped with high-precision instruments. Elevation refers to height above sea level, and inlet height to height above ground level.

| Acronym | Name | Longitude | Latitude | Elevation (m) | Inlet height (m) | Instrument |
|---|---|---|---|---|---|---|
| and | Andilly | 2.3018 | 49.0126 | 175.0 | 60.0 | high-precision |
| cds | Cité des Sciences | 2.3880 | 48.8956 | 43.0 | 34.0 | high-precision |
| cou | Coubron | 2.5680 | 48.9242 | 126.0 | 30.0 | high-precision |
| gns | Gonesse | 2.4205 | 49.0052 | 81.0 | 36.0 | high-precision |
| jus | Jussieu | 2.3561 | 48.8464 | 38.0 | 30.0 | high-precision |
| meu | Meudon | 2.2044 | 48.8025 | 173.0 | 90.0 | high-precision |
| ovsq | OVSQ | 2.0486 | 48.7779 | 150.0 | 20.0 | high-precision |
| rov | Romainville | 2.4225 | 48.8854 | 128.0 | 103.0 | high-precision |
| sac | Saclay | 2.1420 | 48.7227 | 160.0 | 60.0 | high-precision |

et al., 2014). ICON and ART have recently been released under a permissive open source license and are now maintained and developed by a broader consortium of German and Swiss research partners and weather services.

### 2.2.1 Model setup

We ran separate $CO_2$ simulations for the two cities, each nested offline within a larger central European domain. The central European domain was simulated at 6.5 km resolution using grid R3B8 (see Fig. 2), while the inner domains were run at a
resolution of 0.5 km for Zurich (R19B9 grid) and 1 km for Paris (R5B10 grid). Zurich required a higher resolution due to its complex topography and smaller size. The European simulation provided the initial and boundary conditions for the two nested domains and was itself nested into the Copernicus Atmospheric Monitoring Service (CAMS) global inversion-optimized $CO_2$ simulation v24r3, which is based on assimilation of satellite observations (Chevallier et al., 2010; Copernicus Atmosphere Monitoring Service (CAMS), 2020). Meteorological initial and boundary conditions for the European simulation were taken
from the ERA5 global reanalysis of the European Centre for Medium Range Weather Forecasts (ECWMF) (Hersbach et al., 2020). The intermediate European simulation was necessary because of the significant spatial resolution difference between the CAMS and ERA5 global products and the high-resolution simulations for the two cities. To keep the simulated meteorology close to the analyzed meteorology, the ICON-ART meteorological fields were weakly nudged towards the ERA5 data as described in Steiner et al. (2024b).

All simulations were run with 60 vertical layers. The time step was 50 s in the European domain, 10 s for Paris, and 5 s for Zurich. Model output was saved hourly. The Tiedtke-Bechtold convection scheme (`inwp_convection = 1`) was used throughout, but in the city domains, only shallow convection was enabled (`shallowconv_only = .TRUE.`), while in the European domain both shallow and deep convection were active. The configuration for the city domains closely followed the setup of the Swiss weather service for its operational forecasts at 1 km resolution.



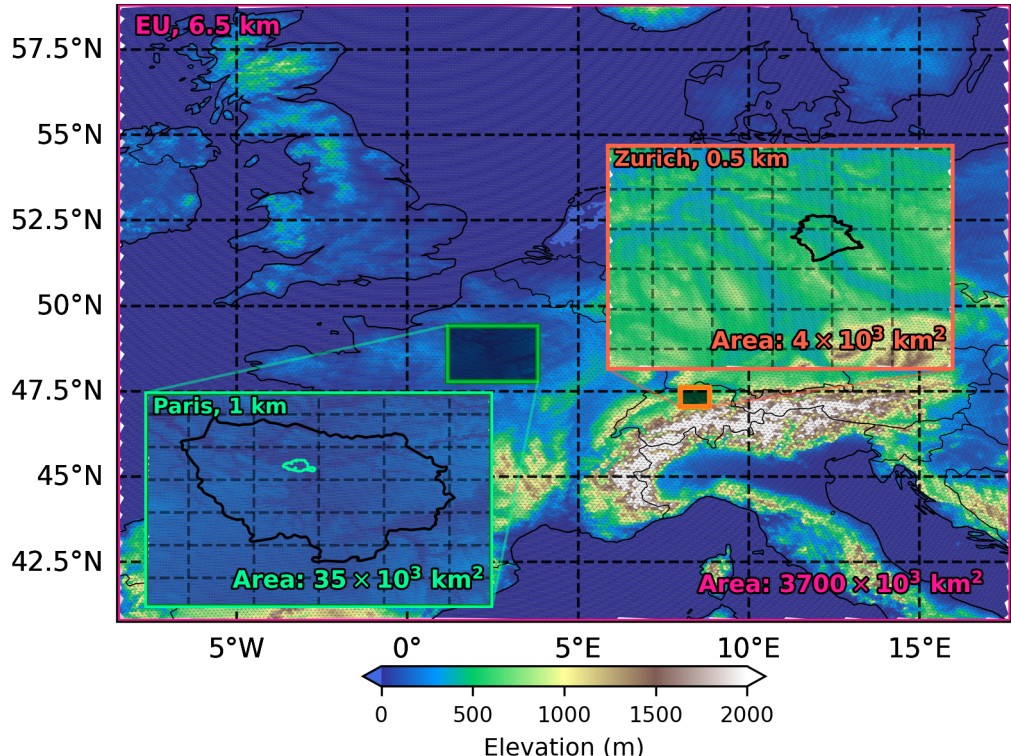

**Figure 2.** Model domains used in this study. The outermost domain covers Central Europe at 6.5 km resolution. The insets show the two nested domains centered on the Zurich and Paris metropolitan areas simulated at a resolution of 0.5 km and 1 km, respectively. The topography is shown as background shading. The black contour denotes the limits of the city of Zurich. For Paris, the black contour corresponds to the Île-de-France, a region covering the agglomerations of Paris. The city limits are represented by the green contour.

Assimilation of $CO_2$ observations was only performed in the two nested domains, not in the European domain. Any potential $CO_2$ mole fraction biases in the European run, which served as boundary conditions for the regional domains, were corrected for by the inversion as described later. To estimate the magnitude of these biases, the simulated mole fractions were compared against measurements from the European Integrated Carbon Observation System (ICOS) (Yver-Kwok et al., 2021). The results will be presented in Sect. 3.1.1.

Different gridded emission inventories were used depending on the simulation domain. For the European domain, the TNO-GHGco inventory (Super et al., 2020) for the year 2021 at about 5 km resolution was used. This inventory relied on the 2023 official reporting of 2021 emissions from the AVENGERS project, except for shipping emissions, which were sourced from the 2021 TNO-GHGco inventory used in the CoCO2 project, reflecting an earlier reporting version for that sector. For the Zurich domain, three inventories were combined: the TNO-GHGco inventory was used for regions outside Switzerland; within

Switzerland, the 2020 Swiss national inventory at 100 m resolution was applied; and for the city of Zurich specifically, this was replaced by a more detailed 2020 inventory as described in Brunner et al. (2025). The city inventory, originally provided as



point, line, and area sources, was rasterized onto the ICON model grid before merging with the other inventories (see Figure 3a). The three inventories were mapped to the ICON grid and merged into a single dataset using the Python package emiproc (Lionel et al., 2025). For the Paris domain, the TNO-GHGco inventory was merged with a 500 m resolution inventory for the

île-de-France (see Fig. 3b) provided by the regional air quality agency AIRPARIF for 2022.

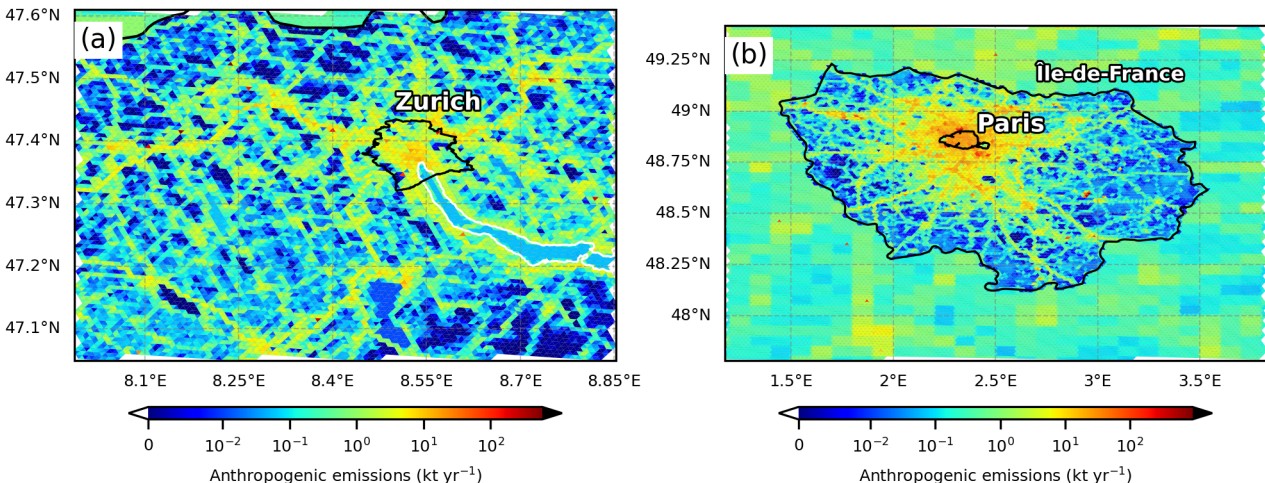

**Figure 3.** Annual mean anthropogenic $CO_2$ emissions in the Zurich (a) and Paris (b) domains in kt yr$^{-1}$. The Zurich data are based on the Swiss national and Zurich inventories, along with TNO-GHGco inventory covering bordering regions of Germany. The Paris data are from AIRPARIF inventory for Île-de-France region, with the remaining area covered by TNO-GHGco inventory. City borders are shown with black lines, and lakes in the Zurich domain are outlined in white.

Biospheric fluxes were computed online using the Vegetation Photosynthesis and Respiration Model (VPRM, Mahadevan et al., 2008) integrated into ICON-ART. As input, it requires shortwave radiation and two-meter temperature, which were provided by ICON, as well as satellite measurements of two indices, the enhanced vegetation index (EVI) and the land surface water index (LSWI) obtained from the MODIS instrument (Vermote, 2015). These indices describe the influence of phenol-

ogy and water stress on photosynthetic uptake of $CO_2$ by vegetation. VPRM independently simulates gross photosynthetic production (GPP) and ecosystem respiration (RE) for seven vegetation classes. The net ecosystem exchange (NEE) is then the difference RE - GPP. The vegetation class-specific parameters of VPRM were taken from Table F1 of Glauch et al. (2025), obtained through an optimization procedure that compared VPRM simulated fluxes with Eddy covariance flux tower observations in Europe.

Another important input for calculating biospheric fluxes is land cover, which provides information about the spatial distribution and type of vegetation in the model domain. We used the CORINE 2018 land cover dataset from the European Environment Agency's Copernicus Land Monitoring Service at 100 m resolution (European Environment Agency (EEA), 2018). However, CORINE fails to resolve small vegetation patches or individual trees in urban areas; thus, a high-resolution vegetation dataset was created for Zurich as described in Brunner et al. (2025) and merged with CORINE data outside the city. Since no such





tailored high-resolution product was developed for Paris, vegetation cover in Paris is likely underestimated. A 10 m resolution version of the land cover map for Zurich is presented in the supplement.

     Land cover in ICON-ART is represented using a tile approach, where multiple land cover types are included in a grid cell proportional to their fractional area. We used the default setting of three tiles, i.e., only the three dominating land cover types were represented. In metropolitan areas, these typically consist of one urban land cover (non-vegetated) and two of the seven

vegetation classes considered by VPRM, often grassland and deciduous forest.

## 2.3    Inversion approach

### 2.3.1    CTDAS inversion framework

To estimate $CO_2$ fluxes, we used the Carbon Tracker Data Assimilation Shell (CTDAS), which is a flexible framework that employs an Ensemble Square Root Filter (EnSRF) to optimize a state vector of flux scaling factors and other elements (Peters

et al., 2005; van der Laan-Luijkx et al., 2017). CTDAS has been coupled with various Eulerian and Lagrangian atmospheric transport models, including ICON-ART (Steiner et al., 2024b). The ensemble approach requires the model to simulate a large ensemble of $CO_2$ tracers, each representing a different perturbation of the state vector elements being optimized.

     Our setup closely followed that in Steiner et al. (2024b). In short, we jointly optimized scaling factors for fluxes and background $CO_2$ mole fractions over individual 1-week assimilation windows. Each week was optimized twice: first with observa-

tions from the current week and a second time with observations from the following week. This assumes that $CO_2$ emitted in the current week also influenced $CO_2$ in the next week.

     Estimating the fluxes for a full year requires 52 individual assimilation cycles, each cycle consisting of three week-long simulations (except for the first cycle, which simulates only two weeks). A schematic illustrating the workflow and information transfer between weeks and assimilation cycles was presented in Steiner et al. (2024b). Figure 4 presents an alternative view

illustrating the ensemble of $CO_2$ mole fractions simulated by the system at an arbitrary observation location.

     The first two weeks comprise a continuous simulation with a restart after the first week to set different flux and boundary condition scaling factors for the second week (see Fig. 4 (a)). ICON-ART provides a restart capability, enabling simulations to be paused, checkpointed, and continued from the same meteorological and tracer fields. Simulated mole fractions are written out hourly and interpolated to station locations using inverse distance weighting, with vertical interpolation based on the two

nearest levels and horizontal interpolation from the five closest ICON cells. Only daytime values (11:00 - 16:00 UTC) are then averaged and used for assimilation. Each week is optimized twice using observations from two weeks, except for the first week of the first cycle (see Fig. 4a) which is optimized only once using the observations from the first two weeks.

     The second cycle starts with another simulation for the first week but now using the optimized scaling factors returned by CTDAS to propagate the optimized boundary conditions and fluxes forward in time to serve as optimized $CO_2$ initial conditions

for the next week. Only one single $CO_2$ tracer is simulated in this case as seen in the left part of Fig. 4b. The second week of the second cycle is then restarted from these optimized initial conditions and a new ensemble is generated based on the once optimized scaling factors for week 2 from the previous cycle. Consequently, the spread among ensemble members in week 2



is smaller in the second cycle compared to the first (compare panels (a) and (b)). The third week restarts from the checkpoint at the end of the second week, generating a new ensemble with a larger spread. Only observations from the third week are assimilated in this cycle, ensuring no observations are assimilated twice. Finally, panel (c) illustrates the third cycle, where the first two weeks are now fully optimized. This process continues for all subsequent cycles, following the same procedure as outlined for the second cycle.

One design choice for the inversion system involved transferring information between different assimilation cycles. One option would be to use the optimized scaling factors from the previous week as prior values for the next week. Alternatively, one could revert to the original prior values, all set to 1. In the first case, the scaling factors optimized for the previous week are assumed to be a good first guess for the present week. In the second case, the weekly scaling factors are assumed to be independent of each other such that each week should restart from the original prior values ($\lambda = 1$). We followed the same approach as in (Steiner et al., 2024b), which blends these two extremes: We computed the new priors as a weighted mean with a weight of 1/3 for the original prior and 2/3 for the posterior from the previous cycle. This approach propagates information from the previous cycle while ensuring that, in the absence of new observations, the fluxes gradually relax back to the prior within a few weeks.

Another aspect that could be transferred between optimization cycles is the state vector uncertainty or ensemble spread. However, we decided to keep the uncertainty for a given cycle unaffected by the optimized uncertainties of previous cycles. Transferring the optimized uncertainties to the next week tends to excessively reduce the ensemble spread. Keeping the value unchanged allows the system to remain stable and adapt quickly to sudden changes in flux intensity or background mole fraction errors.

### 2.3.2 State vector $x$

To estimate anthropogenic and biospheric $CO_2$ fluxes separately, the state vector included separate scaling factors for anthropogenic fluxes, gross photosynthetic uptake (GPP), and total ecosystem respiration (RE). Each factor applies to the fluxes within a group of four neighboring grid cells. This grouping reduces the spatial resolution of the inversion but was necessary to save computation time and memory. In addition, eight scaling factors were included to adjust the background $CO_2$ mole fractions in eight inflow regions following the method outlined in Steiner et al. (2024b). Since an assimilation cycle includes two time windows, each scaling factor is required twice: once for each of the two weeks. For Zurich, the state vector size $s$ is 2 x (2926 x 3 + 8) = 17'572. For Paris, which has a larger domain with 7680 regions, the size was 2 x (7680 x 3 + 8) = 46'126 elements.

The state vector elements are scaling factors $x = (\lambda_1, \lambda_2, .., \lambda_s)$, so the initial prior state vector $x_b$ simply consisted of ones, $x_b = (1, 1, .., 1)$. As mentioned before, from week 3 onwards (see Fig.4), the prior values were defined by a weighted mean between the initial prior and the posterior $\lambda_i$ values from the previous week.



**Figure 4.** Schematic representation of CTDAS assimilation cycles (a) 1, (b) 2 and (c) 3. Black dots indicate hourly daytime (11:00–16:00 UTC) observations which were averaged daily and assimilated during each cycle, while gray dots show all other observations. Each panel shows observations only for days assimilated in the corresponding cycle. $\lambda_i$ denotes the state vector parameters (flux scaling factors) for cycle i, $\lambda_i^{a1}$ and $\lambda_i^{a2}$ indicate once and twice optimized scaling factors, respectively, following the notations in Peters et al. (2005) and Steiner et al. (2024b).





### 2.3.3 Ensemble Square Root Filter approach

In order to optimize fluxes of $CO_2$, the following Bayesian cost function $J$ of observation-model residuals and differences between prior and posterior fluxes is minimized (Peters et al., 2005):

$$J(\boldsymbol{x}) = \frac{1}{2}(\boldsymbol{x} - \boldsymbol{x}^{\mathrm{b}})^{\mathrm{T}}\mathbf{P}^{-1}(\boldsymbol{x} - \boldsymbol{x}^{\mathrm{b}}) + \frac{1}{2}(\mathbf{H}\boldsymbol{x} - \boldsymbol{y}^{\mathrm{o}})^{T}\mathbf{R}^{-1}(\mathbf{H}\boldsymbol{x} - \boldsymbol{y}^{\mathrm{o}}) \tag{1}$$

where $\boldsymbol{y}^{\mathrm{o}}$ denotes the observed mole fractions, $\mathbf{H}$ the observation operator (implemented via the ICON-ART model sampled at station locations), $\boldsymbol{x}$ the state vector (flux and background scaling factors), $\boldsymbol{x}^{\mathrm{b}}$ the "background" (or prior) state vector, $\mathbf{H}\boldsymbol{x}$

the simulated mole fraction, $\mathbf{P}$ the prior error covariance matrix, and $\mathbf{R}$ the observation error covariance matrix. The optimal posterior solution is

$$\boldsymbol{x}^{\mathrm{a}} = \boldsymbol{x}^{\mathrm{b}} + \mathbf{P}\mathbf{H}^{T}\left(\mathbf{H}\mathbf{P}\mathbf{H}^{T} + \mathbf{R}\right)^{-1}\left(\boldsymbol{y}^{\mathrm{o}} - \mathbf{H}\boldsymbol{x}^{\mathrm{b}}\right). \tag{2}$$

CTDAS implements an Ensemble Square Root Filter (EnSRF, Whitaker and Hamill, 2002), or an Ensemble Square Root Smoother if more than one window per cycle is used, to solve this problem. A detailed description of the theory and imple-

mentation of EnSRFs for inverse modeling is provided by Thanwerdas et al. (2025).

In the EnSRF approach, the matrix $\mathbf{P}$ of dimension [$s$ x $s$] is approximated using an ensemble of $N$ state vectors. Let $\mathbf{X}$ denote the ensemble of state vector anomalies of size [$s$ x $N$], representing deviations from the ensemble mean (e.g., the prior has as mean simply 1). Then $\mathbf{P}$ is estimated as

$$\mathbf{P} = \mathbf{Z}\mathbf{Z}^{T} \approx \frac{1}{N-1}\underbrace{\mathbf{Z}\mathbf{G}}_{\mathbf{X}}\underbrace{\mathbf{G}^{T}\mathbf{Z}^{T}}_{\mathbf{X}^{T}}, \tag{3}$$

where $\mathbf{Z}$ is the lower Cholesky decomposition, i.e., one possible definition of the matrix square root, and $\mathbf{G}$ is a Gaussian random matrix of reduced size [$s$ x $N$] drawn from a standard normal distribution (and $\mathbf{G}\mathbf{G}^{T}/(N-1) \approx \mathbf{I}$). This provides a low-rank approximation of the full covariance matrix, such that the problem becomes tractable for high-dimensional inverse problems, as we can now re-write eq. (2) as

$$\boldsymbol{x}^{\mathrm{a}} \approx \boldsymbol{x}^{\mathrm{b}} + \frac{\mathbf{X}\mathbf{Y}^{T}}{N-1}\left(\frac{\mathbf{Y}\mathbf{Y}^{T}}{N-1} + \mathbf{R}\right)^{-1}\left(\boldsymbol{y}^{\mathrm{o}} - \mathbf{H}\boldsymbol{x}^{\mathrm{b}}\right), \tag{4}$$

where $\mathbf{Y} = \mathbf{H}\mathbf{X}$, denotes the ensemble of $N$ modeled mixing ratio deviations from the ensemble mean sampled at the observation locations and times. All these mixing ratios were simulated as separate $CO_2$ tracers within the same forward ICON-ART run. For our inversions, we chose $N$=186, following Steiner et al. (2024b).

### 2.3.4 Observation error covariance matrix $\mathbf{R}$

Uncertainties in the differences between modeled and observed mole fractions are described using the observation error co-

variance matrix $\mathbf{R}$. These uncertainties are caused by measurement errors, transport model errors, and representation errors. Representation errors occur when a model with limited spatial resolution cannot resolve measurements at a single point. In



most cases, this matrix is considered to be diagonal, meaning that each model-observation difference is statistically independent from the others. This is a valid assumption when assimilating daily mean mole fractions from stations separated by large distances. In the case of urban scale inversions with a large number of stations being within 1–5 km from each other, however, this approximation might be not be justified. Several different approaches to account for correlated uncertainties in the $\mathbf{R}$ matrix have been presented, for example, by (Ghosh et al., 2021). Neglecting such correlations may give too much weight to the observations, giving undue weight to the second term of the cost function in Eq. 1.

Due to the high computational cost of full EnSRF inversions, complex formulations of the observation error covariance matrix $\mathbf{R}$, including off-diagonal elements to account for correlated uncertainties between individual observation sites, were only tested in short sensitivity experiments or by re-optimizing individual assimilation cycles without re-running the transport model. These limited tests showed only a small impact on the flux estimates. Therefore, the final inversions presented here employed a diagonal $\mathbf{R}$ matrix (see Eq. 5). Observations are assimilated serially, leveraging the diagonal structure of the observation error covariance matrix $\mathbf{R}$ by processing each observation one at a time.

To estimate the diagonal elements, we first calculated centered (i.e., bias-corrected) weekly average root mean square errors (RMSEs) between modeled and observed mole fractions at each station (Fig. 5) and then applied a centered 5-week moving average. The RMSEs were generally larger in Zurich than in Paris because most of the stations were mid-cost sites measuring above rooftop with high sensitivity to surrounding emissions. The largest RMSEs occurred in the period from November to March and the lowest in late summer. Additionally, elevated RMSEs were observed in the first week of April 2023. In Paris, the highest RMSEs were found for the sites gns, cds and jus. This was likely because cds and jus were located within the city, while gns had a data gap during the late summer months, when model-observation differences are typically smaller.

The smoothed weekly uncertainties for each station served as the model-data mismatch (MDM) thus generating the $\mathbf{R}$ matrix as follows:

$$\mathbf{R} = \begin{bmatrix} MDM_{11}^2 & \cdots & 0 \\ \vdots & \ddots & \vdots \\ 0 & \cdots & MDM_{nn}^2 \end{bmatrix}, \tag{5}$$

where $n$ is the number of observations assimilated in a given cycle.

### 2.3.5 Prior error covariance matrix P

The prior error covariance matrix $\mathbf{P}$ describes uncertainties related to prior flux and background scaling factors, as well as their correlations. These correlations effectively reduce the degrees of freedom, which is necessary to avoid overfitting. The observation networks often lack the density to independently constrain all state vector elements. The length scale of the spatial correlations determines the typical size of structures that can be estimated independently by the inversion. Correlated uncertainties also determine how uncertainties are reduced when averaging over larger domains. For example, if the correlation length exceeds the city size, averaging over the city's grid cells does not reduce uncertainty in total urban emissions.



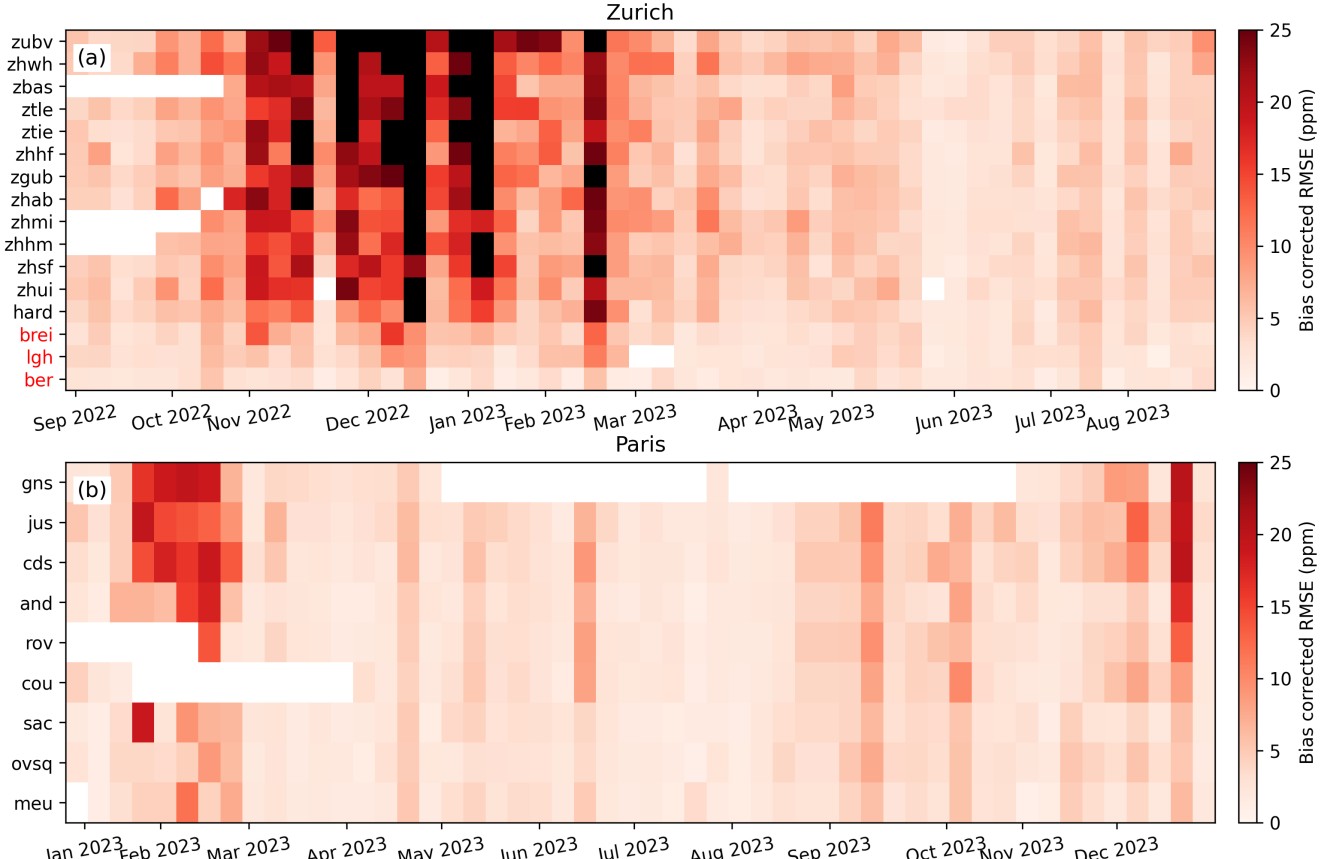

**Figure 5.** Bias-corrected weekly RMSEs in parts per million (ppm) between modeled and observed $CO_2$ mole fractions for each station in (a) Zurich (September 2022–September 2023) and (b) Paris (January 2023–January 2024). Background sites in Zurich are indicated in red on the y axis.

The structure of $\mathbf{P}$ was standardized for both inversions to a common block-diagonal format,

$$
\mathbf{P} = \begin{bmatrix} f_a \cdot e^{-\mathbf{D} \oslash \mathbf{L}_a} & \mathbf{0} & \mathbf{0} \\ \mathbf{0} & f_b \cdot e^{-\mathbf{D} \oslash \mathbf{L}_b} & \mathbf{0} \\ \mathbf{0} & \mathbf{0} & \mathbf{P}_{bg} \end{bmatrix}, \tag{6}
$$

where $e^{\cdot}$ is implied to be taken element-wise, and we use definitions

– $f_a = 1$, $f_b = 0.5$: the prior uncertainty for the anthropogenic and biospheric categories, respectively. Factors between 0 and 1 correspond to standard deviations of 0 and 100 % per flux region,

– $\mathbf{D} \in \mathbb{R}^{N_r \times N_r}$: distance matrix with entries $d_{ij}$ between flux regions $i$ and $j$,

– $\mathbf{L}_a, \mathbf{L}_b \in \mathbb{R}^{N_r \times N_r}$: length scale matrices for the anthropogenic and biospheric categories with entries $L_{ij,a}$ and $L_{ij,b}$,





    – ⊘: element-wise division,

– $\mathbf{P}_{\mathrm{bg}} \in \mathbb{R}^{8\times8}$: block corresponding to background parameters. It has a banded structure with diagonal elements set to $0.005^2$, corresponding to 0.5 % variance per background inflow region, and off-diagonal correlations between neighboring regions: $\pm1$ and $\pm2$ positions are weighted by factors of 0.5 and 0.2, respectively. The total uncertainty is approximately 1 %.

For Paris, the spatial correlation between flux scaling factors was fixed to length scales of $L_{ij,\mathrm{a}} = L_\mathrm{a} = 20\,\mathrm{km}$ and $L_{ij,\mathrm{b}} =$
$L_\mathrm{b} = 50\,\mathrm{km}$ for anthropogenic and biospheric fluxes, respectively. The greater length scale for biospheric fluxes arises from the assumption that vegetation responds to environmental drivers in a correlated manner across neighboring regions in the model domain.

In Zurich, spatially varying correlation lengths were considered. Shorter correlation lengths were applied within the city, as the dense observation network can resolve flux gradients. Longer correlation lengths were utilized outside the city. The spatial
dependence of $L_{ij}$ was defined as a linear function of the average distance between flux regions $i$ and $j$ to the city center $d_{c,ij}$:

$$L_{ij,\mathrm{a}} = \left(\frac{d_{c,ij}}{d_\mathrm{max}}\right)L_\mathrm{a} + \left(1 - \frac{d_{c,ij}}{d_\mathrm{max}}\right)L_{\mathrm{urban,a}}, \tag{7}$$

(and analogously defined $L_{ij,\mathrm{b}}$) where $L_\mathrm{a}$ and $L_{\mathrm{urban,a}}$ are the correlation length limits between which we would like $L_{ij,\mathrm{a}}$ to vary outside and inside the city, and $d_\mathrm{max}$ is the maximum distance between flux regions. Using the mean distance $d_{c,ij}$ was important to preserve the symmetry of the $\mathbf{P}$ matrix. As in Paris, we set $L_a = 20\,\mathrm{km}$ for anthropogenic and $L_b = 50\,\mathrm{km}$ for
biospheric fluxes, while we used $L_{\mathrm{urban,a}} = 5\,\mathrm{km}$ for anthropogenic and $L_{\mathrm{urban,b}} = 8\,\mathrm{km}$ for biospheric fluxes.

The prior and posterior flux uncertainties presented in the Section 3 were computed from the full covariance matrices to account for spatial covariances, similar to the approach described in Steiner et al. (2024b). For each weekly inversion cycle, we calculated:

$$U = \sqrt{\boldsymbol{g}^T \mathbf{P} \boldsymbol{g}}, \tag{8}$$

where $\mathbf{P}$ is the prior or posterior error covariance matrix and $\boldsymbol{g}$ is a vector of 0 and 1 (or a binary mask) corresponding to the selected region of model grid (e.g., selected grid cells inside Zurich, Paris or Île-de-France region). For seasonal or annual means, weekly uncertainties were aggregated using standard error propagation.

## 3 Results

### 3.1 Evaluation of the forward simulations compared to observations

### 3.1.1 European simulation

The main purpose of the European simulation was to provide initial and boundary conditions for the two nested domains. Comparisons with measurements in Europe thus provide information on the magnitude and temporal dynamics of errors in these boundary conditions.





Overall, the mean daily afternoon $CO_2$ mole fractions from the European simulation showed very good agreement with

measurements from the ICOS network. The list of sites included in this comparison is provided in the supplement Fig. S2.

Figure 6 presents monthly mean mole fractions, model biases, RMSEs, and correlation coefficients averaged across all sites.

The annual mean bias was about 1.24 ppm and the RMSE about 4.19 ppm.



**Figure 6.** Monthly statistics of mean modeled versus observed afternoon $CO_2$ mole fractions at ICOS stations in the European model domain. Panel (a) shows the contributions of the different components (background, anthropogenic and biospheric) to total $CO_2$ mole fractions. Total and background mole fractions are offset by –410 ppm. The remaining panels present (b) monthly mean biases, (c) RMSEs, and (d) Pearson correlation coefficient (r).

The high monthly correlation coefficients (mean value of 0.73) suggest that the model captured most of the day-to-day variability in the observations. However, the error statistics show significant variations between months. For instance, biases

in simulated $CO_2$ mole fractions exhibit a clear seasonal pattern. The largest biases, reaching up to 3 ppm, occurred in August and September. This overestimation is likely related to the challenge in accurately modeling biospheric $CO_2$ fluxes, which dominate during the warm season. Photosynthetic uptake and respiration processes introduce uncertainties in the prior fluxes





that are difficult to constrain at the European scale. RMSEs were quite stable over the year with a tendency of higher values during the cold season (October–March). This increase may be associated with uncertainties in residential heating, which is one

of the main anthropogenic sources. Another important factor that may lead to larger errors during the cold season is stronger vertical stratification and shallower boundary layers. The lowest correlation of 0.5 occurred in April 2023. This could be related to transition-season effects, where shifts in both biospheric activity (onset of vegetation activity) and heating demand (end of heating period) introduce additional uncertainty. All correlations were calculated as Pearson correlation coefficient r using the np.corrcoef function in Python.

**3.1.2  Nested simulations over Zurich and Paris**

A similar comparison of the prior simulation against measurements was conducted for the two nested simulations using instead the measurement networks in Zurich and Paris (two stations of the Paris network are also ICOS stations). The results are presented in Fig. 7 (blue bars) in terms of statistics per station rather than per month. To be comparable with the results for the European domain, correlation coefficients were first averaged by month and then averaged over all months. Otherwise,

high correlation coefficients would be obtained due to the strong seasonal cycle in $CO_2$, which the model is usually able to capture well. RMSEs and biases of the prior simulation (blue bars) were generally much larger for stations in Zurich than in Paris. This is due to the different network types with mostly rooftop mid-cost sensors in Zurich as opposed to high-precision measurements on tall towers in Paris. The sites in Zurich are mostly located inside the city and at a much lower altitude above the surface than in Paris. As a result, they are more sensitive to emissions from the city and therefore also to errors in these

emissions. For the two background sites, Beromünster and Lägern-Hochwacht in the Zurich domain, the model showed very similar performance as for the sites in the Paris domain.

The biases and RMSEs in the Paris domain were of similar magnitude to the values of the comparison with ICOS sites in the European simulation. The correlation coefficients were also comparable for both simulations, with Paris showing only a slight improvement in the priors, indicating that higher resolution did not substantially increase correlation in this case.

**3.2  Inversion results**

The inversion significantly improved simulated $CO_2$ mole fractions compared to observations (orange bars in Fig. 7). Daily RMSEs averaged across all stations decreased in both cities, dropping from 11.83 to 7.1 ppm in Zurich, and from 5.33 to 4.25 ppm in Paris. The bias was reduced to near zero, from 2.62 to 0.05 ppm in Zurich and from 1.36 to -0.11 ppm in Paris. The reduction in the biases was mostly a result of the adjustment of the background scaling factors rather than the flux scaling

factors. Finally, the correlation coefficients increased from 0.73 to 0.83 in Zurich and from 0.76 to 0.83 in Paris. Although the correlations were already high in the prior simulation, the inversion further improved the temporal agreement between the model and observations. After inversion, the distribution of errors across sites was more uniform, suggesting an improvement in spatial $CO_2$ gradients.



**Figure 7.** Daytime (11:00 – 16:00 UTC) mean bias, RMSE, and Pearson correlation coefficient (r) values of $CO_2$ mole fractions before and after inversion between ICON-ART and the observations over Zurich (left panels) and Paris (right panels). Blue bars denote the results for the prior simulation, orange bars for the posterior simulation after optimization.

Figures 8 and 9 show timeseries of weekly prior and posterior flux estimates averaged across Zurich and Paris, as well as
their surrounding agglomerations. For Zurich, the larger region encompasses the entire area of the nested simulation. In Paris, it corresponds to the Île-de-France (black contour in Fig. 2).

In Zurich, most flux adjustments were concentrated within the city, while domain-wide totals remained close to the prior estimates (Figure 8). Anthropogenic emissions, the dominant source of $CO_2$ in the urban area, were reduced in almost all weeks, especially during the heating season. The reductions were particularly large in the last weeks of December 2022 and
the first week of January 2023. This was a period of comparatively warm temperatures and coincided with the Christmas and New Year period when many residents leave the city for holidays. Our prior estimates of emissions from residential





heating accounted for outdoor temperatures by following a heating-degree-days approach (see Brunner et al. (2025) for details). However, the strong reduction in the posterior estimates during warm winter periods suggests that this approach may not accurately capture the influence of temperature. Another factor, which likely contributed to the lower posterior emissions, was the energy crisis driven by the Russo-Ukrainian War. This crisis led to strongly increased prices for gas and electricity. Due to the shortage in primary energy, the Swiss government formulated the goal of a reduction of gas consumption by 15 %, a goal that was exceeded during the heating period October 2022 to March 2023 (https://www.news.admin.ch/de/nsb?id=94439). Our prior emissions did not account for this factor. Despite these plausible factors, the reductions in winter were likely too strong as discussed below in Section 4. The prior emissions showed a maximum in winter and a minimum in summer, which is expected given the important contribution of heating to total $CO_2$ emissions in the city. The posterior fluxes showed a very different behavior that does not seem to be realistic.

The inversion significantly increased the net uptake of $CO_2$ by the biosphere in late May/June 2023, suggesting that prior estimates underestimated photosynthetic activity. In other periods, the posterior NEE fluxes remained closer to the prior. In most weeks, NEE was much smaller than the anthropogenic fluxes in the city of Zurich, but in summer, they were of similar magnitude. The increase in posterior uptake in June was partially offset by higher anthropogenic emissions, suggesting poor separation between anthropogenic and biospheric fluxes during this period.

Some studies, such as Lauvaux et al. (2016); Lian et al. (2021), focused only on the dormant season to minimize such interference from biospheric fluxes when estimating anthropogenic emissions. Comparing uncertainty reductions between summer and winter thus helps assess how biospheric fluxes influence the inversion constraints. For Zurich, the inversion resulted in notably larger reductions for anthropogenic and total flux uncertainties in winter (both about 90 %) than in summer (about 45 and 35 %, respectively). This difference may partly result from the inversion attributing model transport errors to emissions during winter shallow boundary layer conditions, as discussed earlier. In summer, Zurich showed relatively higher uncertainty reduction for respiration fluxes (about 33 %) compared to winter (about 20 %), while the uncertainty of photosynthetic uptake by the plants (GPP) remained largely unchanged in both seasons. The uptake uncertainties remaining high are likely due to the lack of assimilated nighttime observations, which would help separate RE and GPP. Detailed information about summer and winter prior and posterior fluxes is provided in the Table S3 in the Supplement.

In Paris, the inversion produced much smaller adjustments compared to Zurich (Figure 9). The total and anthropogenic $CO_2$ fluxes remained close to the prior estimates, not only in the Île-de-France region, but also in the city. Biospheric fluxes showed minimal changes in both regions, indicating that the inversion system found little evidence to revise the prior estimates. The posterior uncertainties were significantly reduced for anthropogenic fluxes but remained almost unchanged for biospheric fluxes, suggesting that the observation network was less sensitive to $CO_2$ exchange with vegetation. This aligns with the fact that NEE fluxes were nearly ten times smaller than anthropogenic fluxes. This may result from the CORINE land cover dataset, which inadequately resolves vegetation in cities.

Overall, the relatively small updates in Paris might be explained by a better initial agreement between the model and observations, as well as by a lower signal-to-noise ratio, with anthropogenic enhancements being less distinct against background variability at the tall towers in Paris compared to the rooftop measurements in Zurich.

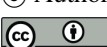



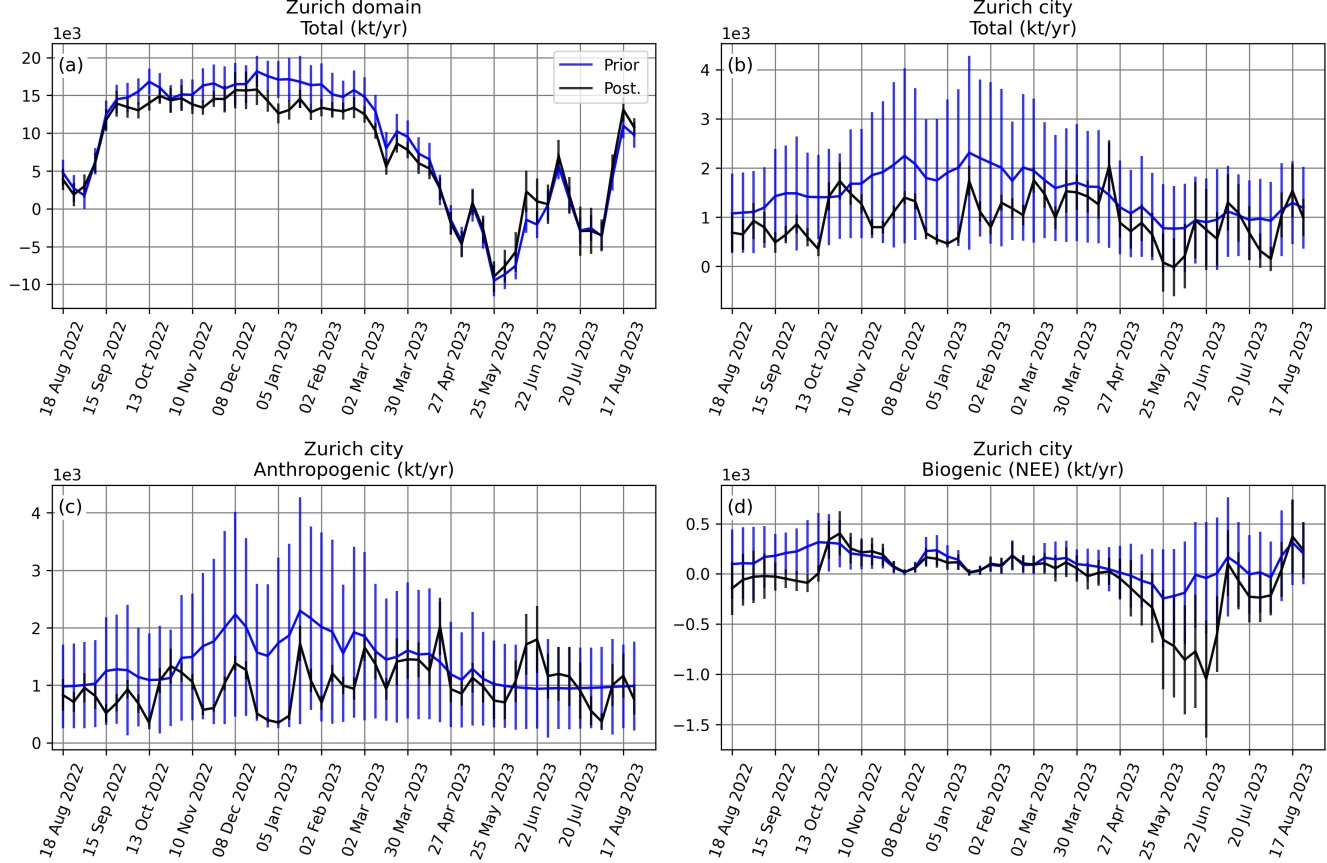

**Figure 8.** Time series of prior and posterior $CO_2$ fluxes in Zurich in (a) the total domain and (b) the city. The remaining panels show the anthropogenic (c) and biospheric (d) $CO_2$ fluxes in Zurich. Biospheric fluxes are shown in terms of net ecosystem exchange (NEE). Vertical bars denote the $1\sigma$ uncertainty in the domain-averaged fluxes computed from the prior and posterior error covariance matrices.

Consistent with this, the inversion barely adjusted biogenic flux uncertainties in Paris (e.g., GPP uncertainty reduction was -0.4 % in summer and -6.0 % in winter). This underlines the importance of integrating high-resolution urban vegetation data to better resolve biogenic fluxes within city domains. In Paris, the reductions in total and anthropogenic flux uncertainties were

similar in both seasons (about 75 - 78 %), with slightly lower reductions in summer, indicating that anthropogenic emissions were effectively constrained owing to the limited sensitivity to biogenic fluxes.

In both cities, the inversion optimized anthropogenic and biospheric fluxes as well as background $CO_2$ mole fractions from eight inflow regions (Figure 10). The Zurich case showed larger changes in background levels and a wider spread between different wind directions. Scaling factors were adjusted by up to 2 %, which corresponds to about 8 ppm given a background

of about 400 ppm. Background mole fractions were consistently scaled down when air was advected from the southern sector



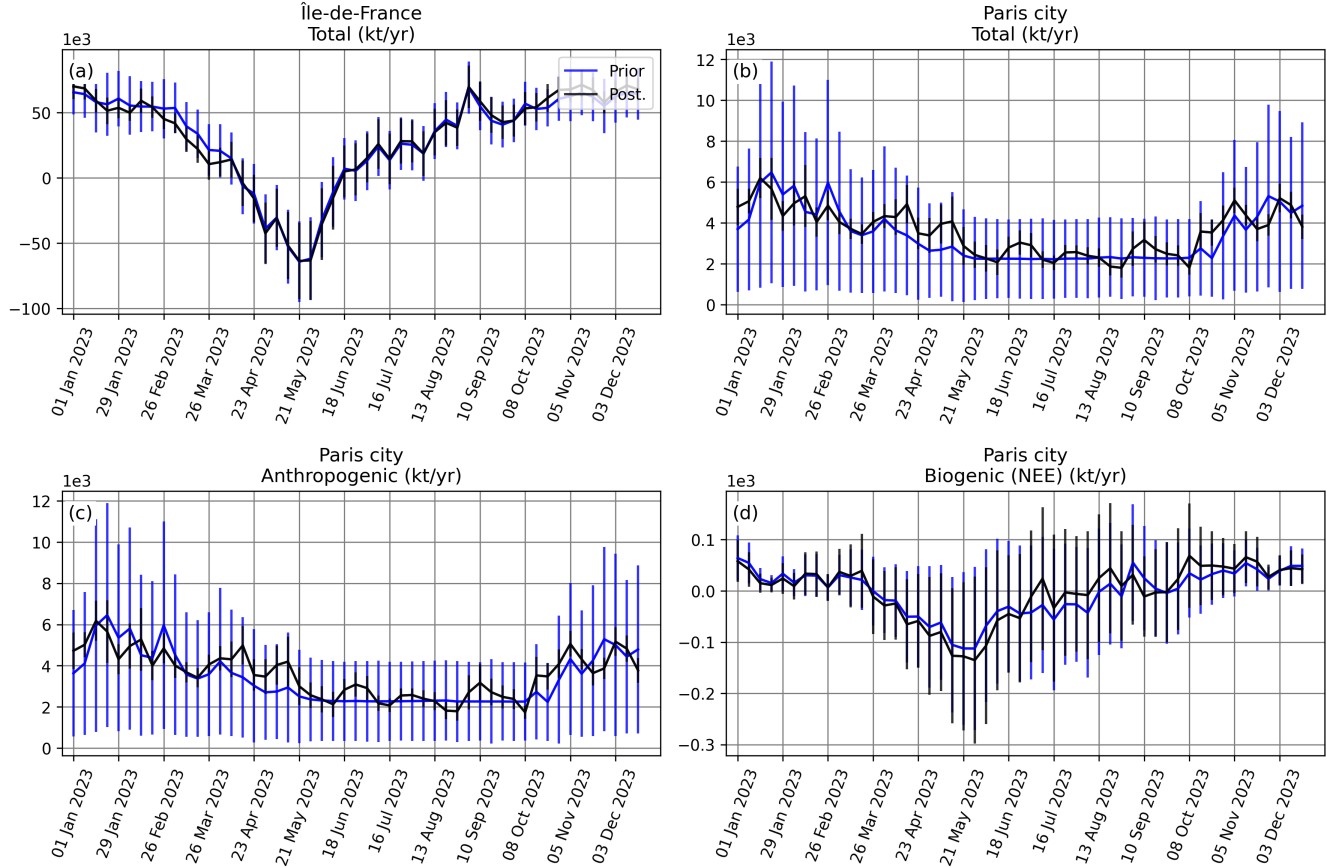

**Figure 9.** Time series of prior and posterior $CO_2$ fluxes in Paris in (a) the Île-de-France region and (b) the city. The remaining panels show the anthropogenic (c) and biospheric (d) $CO_2$ fluxes in the city of Paris. Biospheric fluxes are shown in terms of net ecosystem exchange (NEE). Vertical bars denote the $1\sigma$ uncertainty in the domain-averaged fluxes.

(SSE and SSW) but scaled up when advected from the north (NNE and NNW). The eastern sectors (ENE and ESE) showed upward corrections in winter but mostly downward corrections in summer.

In contrast, the results for Paris exhibited smaller adjustments, mostly within 1 %, and more consistent between the different wind directions, indicating a systematic overestimation in the prior. For both cities, the adjustments are expected to correct

for biases in the boundary conditions provided by the European simulation, which itself is not corrected for biases through data assimilation. The larger corrections for Zurich might be due to the city being more strongly influenced by the European continent due to its position further east from the Atlantic compared to Paris.

Figure 11 summarizes the annual mean $CO_2$ fluxes and their relative changes for Zurich and Paris. The total fluxes are substantially higher in the French capital, reflecting not only its larger area but also its greater population and correspond-

ingly stronger emissions. Net biospheric fluxes (NEE) are only significant for the larger domains (with net uptake in both the





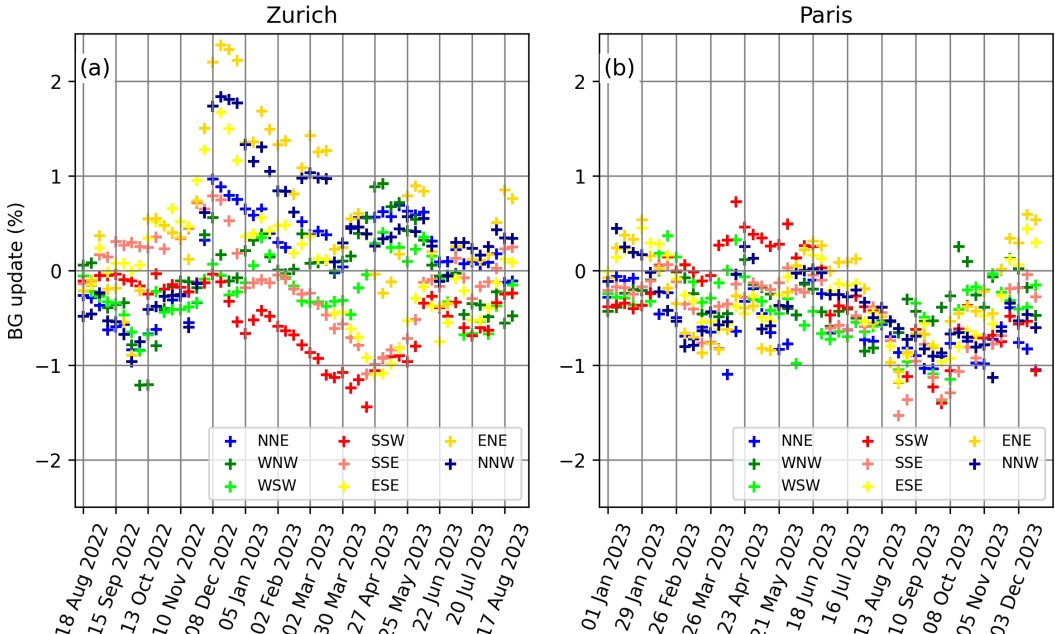

**Figure 10.** Optimized scaling factors for background $CO_2$ mole fractions in (a) Zurich and (b) Paris. The different colors correspond to the 8 different inflow regions.

prior and posterior fluxes) but are negligible over the two cities. The relative changes are thus driven almost entirely by updates to anthropogenic emissions. In Paris, both domain-wide and city-level fluxes remain close to the prior, though total and anthropogenic fluxes increase by about 7 % in the city. In contrast, more substantial adjustments are found for Zurich: domain-wide fluxes are reduced by approximately 10 %, primarily due to large reductions within the city, where annual anthropogenic emissions decrease by 27 %.

The spatial distribution of annual mean anthropogenic $CO_2$ flux updates for both areas is shown in Figure 12. Larger adjustments are evident in Zurich, particularly within and upstream of the city, reflecting the high sensitivity of the observation network to these regions. Emissions were reduced in the eastern and western parts of Zurich, corresponding to residential and vegetated areas, with the largest decrease occurring near the western border, an area dominated by vegetation. In contrast, fluxes in Zurich's center were scaled up. In Paris, updates are generally smaller. The results for the Île-de-France show a slight increase north of the city. This is the region covered by the sites gns and cds, which showed particularly large deviations from the ICON-ART model simulation (see Fig. 5). Vegetated zones to the east and west of Paris show minimal changes in anthropogenic fluxes, as expected. Within the city, a minor redistribution of emissions is visible with reductions in the northern districts and increases in the south.



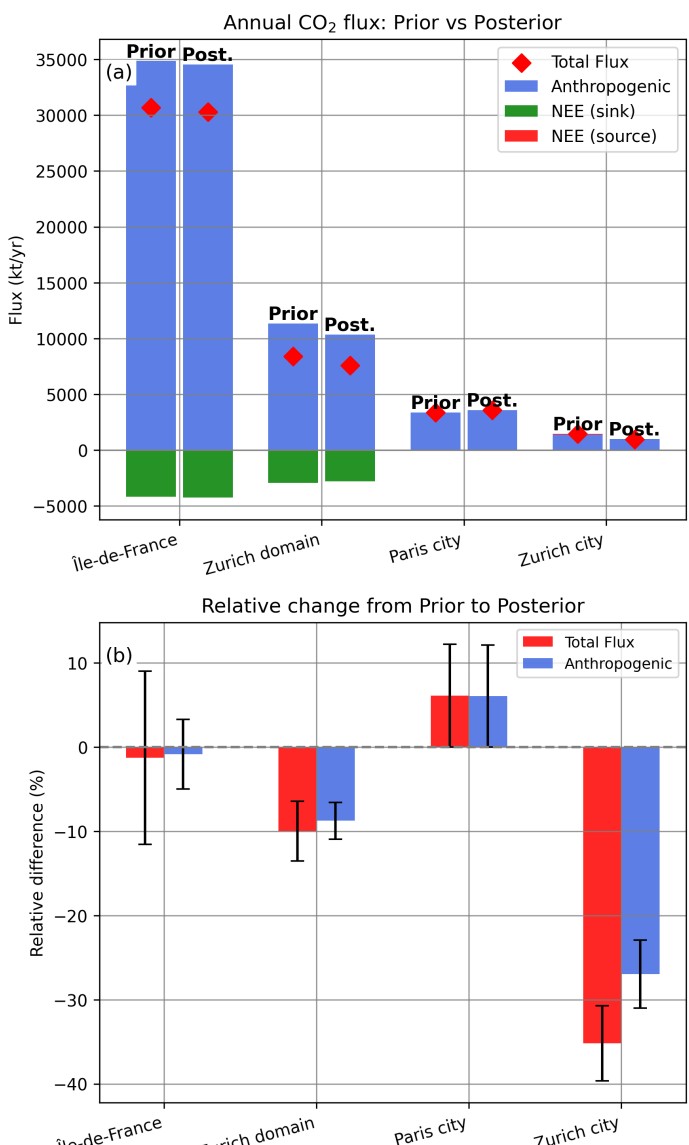

**Figure 11.** Annual total, anthropogenic and biospheric $CO_2$ fluxes in Zurich and Paris (a). Relative changes in total and anthropogenic fluxes (b).

## 4 Discussion

Posterior anthropogenic emissions in Zurich were likely underestimated (i.e., scaled down too much) in some periods, especially in late December 2022 and early January 2023. As shown in Fig. 13 for one of the rooftop sites, the ICON-ART model tended to underestimate wind speeds during periods of low winds. This behavior is consistent with previous studies, which



**Figure 12.** Maps of annual mean anthropogenic $CO_2$ scaling factor updates in (a) the Zurich model domain, (b) the Paris model domain, (c) the city of Zurich and (d) the city of Paris.

showed that mesoscale transport models tend to underestimate wind speed and mixing under stable, low-wind conditions, par-

ticularly in urban environments with enhanced surface roughness (Bréon et al., 2015; Lauvaux et al., 2016; Wu et al., 2018). The selected period, from 15 December to 13 January, coincides with the largest downward adjustments in emissions. When modeled wind speeds exceeded $1\,\mathrm{m\,s^{-1}}$, the model captured the observed day-to-day variability reasonably well. However, during calm episodes, the model often simulated wind speeds below $1\,\mathrm{m\,s^{-1}}$ for several consecutive days, whereas the observations remained higher. These prolonged calm periods result in excessive $CO_2$ accumulation in the model. The inversion system com-

pensates for this overestimation by reducing the emissions. This highlights how errors in representing the actual meteorology, particularly during low-wind episodes, can introduce artifacts into the inversion.

The network-wide statistics presented in Fig. 14 underscore the systematic nature of these transport errors. The figure shows how prior $CO_2$ biases and RMSEs vary with modeled wind speed across all sites in Zurich and Paris. In Paris, where sensors are installed on tall towers well above the surface layer, modeled wind speeds fell below 1 m/s in less than 2 % of observations.




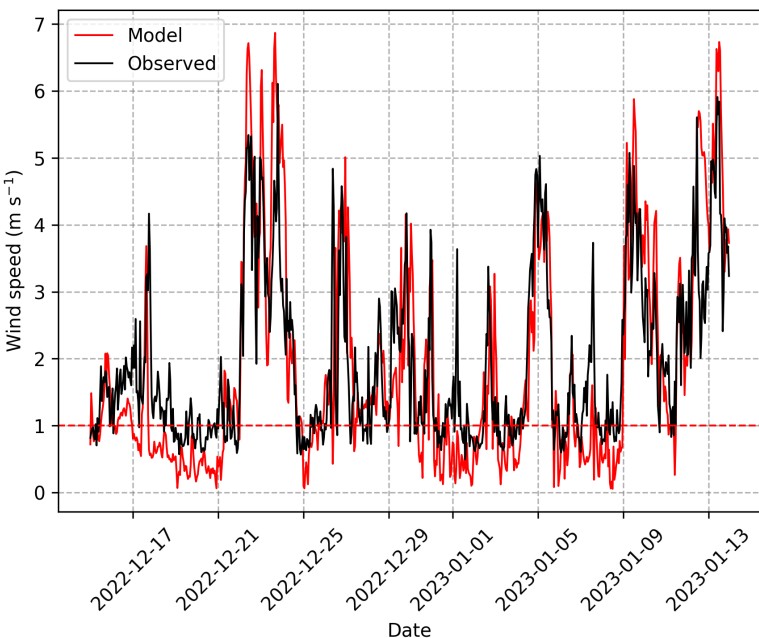

**Figure 13.** Comparison of modeled and observed wind speed at the rooftop site Kantonales Labor in Zurich for the period 15 December 2022 to 13 January 2023.

At the rooftop sites in Zurich, in contrast, wind speeds below 1 m/s occurred during nearly 15 % of all measurement times. This wind speed threshold marks a sharp transition in model performance: Biases in simulated versus observed $CO_2$ mole fractions increase to around 9 ppm, and RMSEs exceed 25 ppm. These statistics reflect a regime in which the model systematically underestimates wind speeds allowing $CO_2$ to accumulate unrealistically. Rather than being isolated events, calm conditions were frequently observed at most sites. The problems of ICON-ART in capturing these situations negatively impact flux estimates during stagnant weather episodes in winter.

The inversion results revealed significant differences in background $CO_2$ mole fraction corrections between Zurich and Paris. The larger background $CO_2$ mole fraction adjustments in Zurich compared to Paris are likely related to the difference in their geographic position. Paris, located closer to the Atlantic Ocean, is exposed to air masses less affected by European anthropogenic emissions. In contrast, Zurich lies further inland and is more influenced by continental sources, resulting in a more complex and variable background signal. This makes it more challenging to differentiate between the influence of local emissions and background changes. Especially in winter when anthropogenic emissions increase not only in Zurich but also across Europe. This is reflected in the background adjustments: during periods of large reductions in anthropogenic fluxes in Zurich, the background $CO_2$ was actually increased for the NNW, ENE, and ESE wind sectors. This pattern is explained by all three background tower sites (Beromünster, Laegern-Hochwacht, and Birchwil Turm). All of them were showing elevated



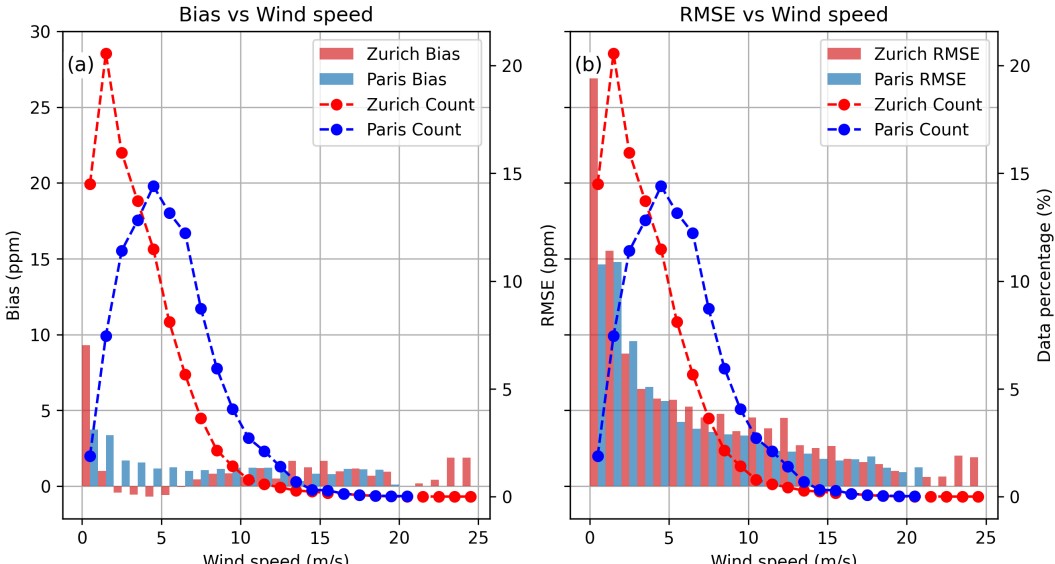

**Figure 14.** Bias and RMSE of $CO_2$ mole fractions as a function of simulated wind speed in Paris and Zurich.

levels of $CO_2$ mole fractions higher than those simulated by the model, indicating a regional-scale increase in emissions that was not fully captured in the prior model setup.

Another limitation of the model involves the VPRM model. In our simulations, VPRM relied on satellite indices derived from MODIS observations at a resolution of 250 to 500 m, which is insufficient for resolving urban vegetation. This poses a challenge when modeling photosynthetic uptake in heterogeneous urban environments. For instance, Ren et al. (2017) conducted a spatio-

temporal analysis using Landsat data and field measurements and demonstrated that mixed-pixel effects in urban settings severely compromise estimates of tree density and structural vegetation attributes, especially in areas with impervious surfaces such as roads. Likewise, (Velasco et al., 2016) emphasize that urban greenery is often poorly captured in models, and argue that no method currently exists to directly evaluate the $CO_2$ uptake by urban vegetation. Together, these studies suggest that current urban biosphere models, which are relying on moderate-to-high-resolution satellite imagery, may systematically underestimate

the contribution of small vegetation patches and street trees to total $CO_2$ fluxes, and misrepresent the net carbon exchange in densely built-up areas. In Paris, this issue was likely exacerbated by using the coarse CORINE land cover dataset at 100 m resolution.

The impact of biospheric flux uncertainties on the estimation of anthropogenic emissions becomes increasingly important during the summer months. In our VPRM simulations for Zurich, prior net ecosystem exchange (NEE) was mostly positive (net

source) in the city center, even in summer, with a mean NEE of +0.29 $\mu$mol m$^{-2}$ s$^{-1}$. In contrast, surrounding vegetated areas functioned as sinks, as expected for the growing season (Figure15). The positive NEE in the city center appears unrealistic, suggesting an underestimation of photosynthetic uptake by urban vegetation, likely due to the coarse resolution of satellite



data. Respiration in VPRM only depends on temperature but not on satellite indices. The different impacts of low resolution MODIS observations on photosynthesis and respiration may thus explain the unrealistic net positive NEE values.

After optimization, the inversion reduced respiration and enhanced uptake, particularly in surrounding green areas, resulting in a mean NEE of –2.15 $\mu$mol m$^{-2}$ s$^{-1}$ averaged across the city. This adjustment brings the posterior fluxes into better agreement with local eddy covariance measurements (Brunner et al., in preparation), which show that daytime photosynthetic uptake during summer can exceed concurrent anthropogenic emissions, leading to net negative fluxes in certain areas.

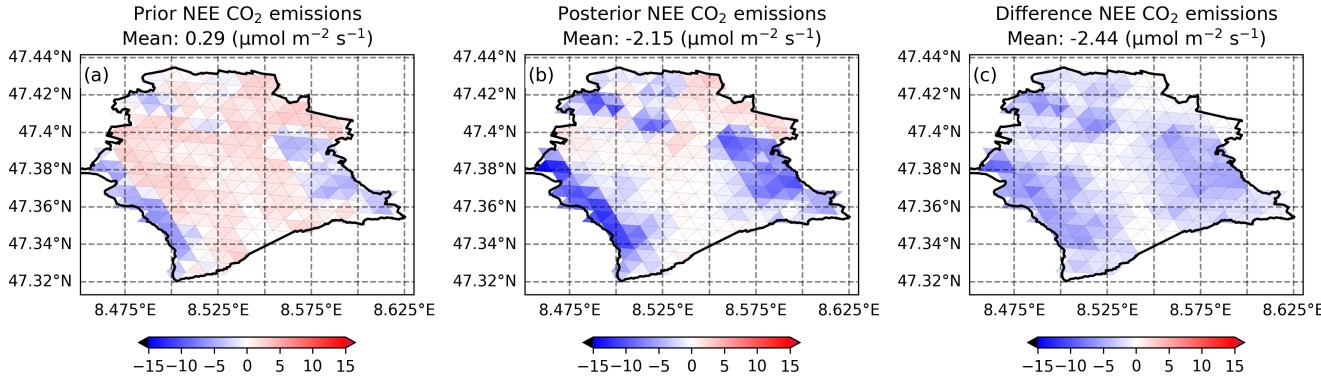

**Figure 15.** Net ecosystem exchange (NEE) fluxes in Zurich in summer (Jun/Jul/Aug) 2023. (a) Prior NEE, (b) posterior NEE, and (c) posterior – prior difference.

Together, these results illustrate how inversion outcomes are influenced by transport model biases and simplifications in 495   prior flux estimates, which vary by space, season, and flux type. The observed corrections provide a quantitative view of where and when the inversion deviates from the prior, revealing limitations in current models and offering guidance for improving urban $CO_2$ flux estimation.

Table 3 summarizes the anthropogenic $CO_2$ emission estimates from this study. It also includes information on the inventory data used as a prior and literature values for Île-de-France and Paris. The inversion reduced prior estimates for Zurich city from 500   $1388.9 \pm 156.8$ to $1012.3 \pm 38.8$ kt yr$^{-1}$, although no independent literature estimate exists for comparison. For Île-de-France, the prior and posterior values are close to each other, and about 16 % lower than the earlier estimate in (Staufer et al., 2016) between August 2010 and July 2011. In Paris, the posterior estimate of $3580.0 \pm 101.9$ is slightly higher than the prior $3375 \pm 172$ kt yr$^{-1}$ and in good agreement with the estimate for the year of 2020 of $3650 \pm 1830$ described in (Nalini et al., 2022). Overall, the inversion reduced posterior uncertainties compared to priors, demonstrating the added value of atmospheric 505   constraints in emission estimates.





**Table 3.** Prior and posterior anthropogenic $CO_2$ emissions in Zurich, Île-de-France region and Paris based on the results from this study, inventory data and other independent estimates.

| Region / City | Source | Year | Anthr. $CO_2$ emissions (kt yr$^{-1}$) | Remarks |
|---|---|---|---|---|
| Zurich | Mapluft | 2020 | $1388.9 \pm 156.8$ | Used as prior |
| | This study | Sept. 22 - Aug. 23 | $1012.3 \pm 38.8$ | Posterior |
| Île-de-France | AIRPARIF | 2022 | $34849.6 \pm 2181.3$ | Used as prior |
| | This study | 2023 | $34553.8 \pm 880.0$ | Posterior |
| | Staufer et.al 2016 | Aug. 2010 - Jul. 2011 | 40900 | |
| | Lian et.al 2023 | 2021 | $34300 \pm 2300$ | |
| Paris | AIRPARIF | 2022 | $3375.1 \pm 429.2$ | Used as prior |
| | This study | 2023 | $3580.0 \pm 101.9$ | Posterior |
| | Nalini et.al 2022 | 2019 | $6410 \pm 2330$ | |
| | Nalini et.al 2022 | 2020 | $3650 \pm 1830$ | |

## 5 Conclusions

In this study, we applied the ICON-ART-CTDAS inversion framework at high spatial resolution to estimate urban $CO_2$ fluxes in Zurich and Paris over a full annual cycle. Simulations were conducted for a central European domain along with nested domains centered on each city. To better capture the influence of complex terrain on mesoscale flow dynamics, the Zurich
domain was simulated at a finer resolution (500 m) compared to Paris (1 km).

Forward simulations were evaluated against $CO_2$ observations in all three domains. Simulated mole fractions for the European domain showed high (monthly mean) correlations with observations from the ICOS network, indicating that the model captures effectively day-to-day variability. Biases and RMSEs were of the order of a few ppm. Additionally, the model reproduced daily variations in wind speed and temperature well (see Supplementary Information). However, comparisons revealed a
strong sensitivity to wind speed, with mole fraction errors increasing substantially during calm periods, limiting the inversion performance in Zurich with its complex terrain. In contrast, the Paris domain, characterized by flatter terrain and a tower-based measurement network, exhibited fewer instances of low wind speeds and overall smaller model–observation mismatches.

In Zurich, the inversion reduced anthropogenic emissions by 27 % relative to the prior inventory, resulting in a posterior annual emission of $1012.3 \pm 38.8$ kt yr$^{-1}$, whereas in Paris, emissions increased by 7 %, yielding a posterior emission of
$3580.0 \pm 101.9$ kt yr$^{-1}$. These differences do not necessarily indicate varying accuracy in the inventory estimates but are partly due to factors such as observational constraints. Zurich's dense rooftop sensor network provides greater sensitivity to local sources, while Paris' tower-based measurements better represent the regional boundary layer. In Zurich, seasonal patterns revealed a clear anti-correlation between optimized anthropogenic emissions and 2-meter air temperature, highlighting the significant contribution of residential heating to wintertime $CO_2$ emissions. The most significant reductions appeared





under stagnant weather with low wind speeds and shallow mixing layers, causing inflated simulated $CO_2$ concentrations and necessitating stronger downward flux corrections. Even minor changes in emissions during these periods produced large modeled enhancements in urban concentrations, indicating the inversion's sensitivity to transport errors. In contrast, seasonal corrections in Paris were more moderate. The higher sensor placement and fewer calm periods reduced the occurrence of artificial $CO_2$ accumulation in the model, limiting the magnitude of such inferred flux changes.

Our results also underscore the importance of accurately representing prior biospheric fluxes in urban areas, especially in summer when vegetation uptake significantly influences total $CO_2$ fluxes. In Zurich, prior estimates of net ecosystem exchange (NEE) indicated an unrealistic net carbon source in the city center. The inversion corrected this by reducing respiration and enhancing uptake in vegetated areas, aligning posterior fluxes more closely with local eddy covariance observations. This highlights the need for improved representation and parameterization of urban vegetation in biospheric flux models to better 535 capture seasonal and spatial variability.

We did not perform any meteorological data assimilation and did not have access to a meteorological ensemble, which limited our ability to explicitly quantify transport-related uncertainties in the simulated $CO_2$ mole fractions. To estimate model–data mismatch (MDM), we used a pragmatic approach based on bias-corrected RMSE values calculated for each station. Periods of large model–observation discrepancies, especially under low wind speeds, resulted in exaggerated emis-540 sion corrections, as the inversion system attempted to compensate for transport biases through flux adjustments. We addressed this issue by rejecting extreme outliers. Although this improved the stability of the inversion, addressing transport related biases more fundamentally will require improved meteorological input, potentially through the use of transport ensembles (e.g., Steiner et al. (2024a); McNorton et al. (2020)) or joint optimization of fluxes and meteorology.

As urban $CO_2$ flux estimation becomes increasingly central to net-zero planning, improving both transport and biospheric 545 model fidelity, as well as integrating enhanced observational constraints, will become essential for reliable emissions monitoring in cities. Projects like ICOS Cities (Lan et al., 2024) have demonstrated the importance of deploying tall-tower eddy covariance, isotopic, and street-level sensors in urban environments to validate emission inventories and improve model constraints on anthropogenic and biospheric fluxes. The CoCO$_2$-MOSAIC 1.0 dataset (Urraca et al., 2024) is complementary to these observational efforts and offers high-resolution ( 0.1°) emission priors, which is in line with the crucial role of detailed 550 bottom-up information in urban inversion frameworks.

Overall, our study highlights the sensitivity of urban $CO_2$ flux inversions to transport model biases, observational network design, and uncertainties in prior flux estimates. Addressing these challenges requires integrated approaches that combine high-resolution meteorology, improved biospheric flux modeling, and expanded observational networks tailored to urban complexity. Future research should explore joint inversion frameworks that incorporate both meteorology and flux uncertainties and 555 leverage transport ensembles to better constrain emissions. These advances will be critical for reliable urban carbon monitoring and for supporting policy efforts aimed at reducing emissions in complex city environments.



*Data availability.* The observation data used in this work are described and available via the ICOS Carbon Portal (`https://icos-cp.eu/`, accessed 27 July 2025) and the ICOS Cities data portal, respectively (`https://citydata.icos-cp.eu/portal`, accessed 27 July 2025). The CORINE land cover data is available via the Copernicus Land Monitoring Service (`https://land.copernicus.eu/en/products/corine-land-cover`, accessed 27 July 2025). The ICON model is an open-source modeling framework for weather, climate, and environmental prediction, available under DOI: 10.35089/wdcc/iconrelease2025.04 (ICON partnership, 2025). ICON-ART, the atmospheric chemistry and aerosol module, is developed at KIT and included in this release for research purposes. The specific ICON-ART model branch used in this study is called icon-kit-empa-np and is available upon request from the main developers. Access to the Git repository containing the official CTDAS code is available upon request from the main developers.

*Author contributions.* NP prepared the inversion framework, designed the setup, performed and analyzed the simulations, and prepared the manuscript, figures and tables. DB supervised the study, supported the design of the simulation experiments, and contributed to the manuscript writing. MS and EK contributed to the development of the inversion framework and the model code and provided valuable input through frequent discussions. LE and MR led the conception, coordination, and implementation of the measurement networks in Zurich and Paris, respectively, and contributed to the overall study design. PR installed and maintained the instrumentation in Zurich. SG worked on the calibration and observation data processing pipeline, evaluated sensor performance, and performed the initial data quality control. LC and LD provided emission inventory data for the nested Zurich and Paris city model domains. LC also contributed to the preparation of the land use input data. All authors contributed to the interpretation of the results and the manuscript revision.

*Competing interests.* The authors declare no competing interest.

*Acknowledgements.* This work was funded by the European Union's Horizon 2020 research and innovation programme, grant agreement number 101037319, named Pilot Applications in Urban Landscapes - towards integrated city observatories for greenhouse gases (PAUL) known as ICOS Cities. The ICON-ART inversions were conducted at the Swiss National Supercomputing Centre (CSCS) under grant No. s1302 and were supported by the Center for Climate Systems Modeling (C2SM). The authors would also like to thank the ICOS station PIs for providing $CO_2$ dry air mole fractions for model evaluation. The authors are also grateful to Junwei Li for his significant contribution to the preparation of the land use data used in this study. Finally, we would like to thank the whole modelling team from Munich, led by Jia Chen, for the regular meetings, fruitful discussions, and valuable insights that greatly supported this work. We acknowledge the use of automated grammar and punctuation tools provided by Overleaf. In addition, we used AI-assisted language support to improve phrase clarity in certain cases.



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
