# Peer review of "Estimation of CO2 fluxes in the cities of Zurich and Paris using the ICON-ART CTDAS inverse modelling framework"

_EGUsphere, 2025_

## Referee Comment (RC2)

**Review of Ponomarev et al. (2025)**

This study presents an investigation of urban CO2 fluxes in Zurich and Paris through inverse modeling. It is interesting that the same modeling framework is applied to two different cities with different sizes, emission distributions and set-ups of their measurement network. However, I have some concerns regarding the set-up of the inversions, which are mostly mentioned in the paper, but not addressed in any type of sensitivity tests. I appreciate that the study covers a lot of ground, making it hard to dig deep everywhere. However, in the absence of sensitivity tests, the final emission estimates need to be presented more carefully. They should be accompanied by a more comprehensive discussion of the limitations and uncertainties, with suggestions on how these can be addressed in future work.

**My primary concerns:**

1. I think the authors undersell how complex their inversion set-up is compared to previous urban work. The system co-optimizes anthropogenic emissions and GPP and RE and boundary conditions. Previous work has shown that co-optimizing anthropogenic emissions and NEE with only CO2 observations is already difficult, if not situationally impossible (e.g., Sargent et al., 2018). This is one of the key difficulties in urban inversions. Yet, in most of the manuscript, the authors consider that all these components can be separately estimated, which is something that, to my knowledge, has never been shown before. It would therefore be important that the authors address how good their estimates are. The authors do address that the biosphere fluxes are likely too low in the prior, which makes co-optimization easier. Then, however, I think it would be an easy sensitivity test to run the inversion with biosphere prior x2 or x5. Additionally, given the inversion method, the authors can investigate the posterior covariances to see if estimation of GPP, RE and anthropogenic emissions are correlated. The current manuscript does not include analyses of this novel aspect of the study. I would

As a specific example, I find it hard to believe that separate estimation of RE and GPP is possible, and that the interpretation of these two separately is meaningful (L490-493), especially with only afternoon observations. My hypothesis would be that the posterior covariances will indicate that the solutions for RE and GPP are correlated, and if this is not the case I would be very interested in the authors' interpretation of this result.

really like to see some analysis of posterior covariances between flux categories,

which is possible without additional inversions.

2. With the lack of sensitivity tests, the authors rely on the posterior error statistics. While these have value, they are very optimistic as absolute uncertainties. I think the authors should be much more explicit that these statistics provide a limited

view of the real uncertainties, which generally require sensitivity tests to be quantified. Reading the abstract would give the reader the (in my view false) impression that there are tiny posterior uncertainties and the system works incredibly well. In reality, many major uncertainties are not included in the current uncertainty estimate. In fact, my interpretation is that for Zurich the authors find that in winter the estimate is biased due to transport errors, and in summer the estimate is biased due to interference from the biosphere. Lian et al. (2023) also report a 3x higher posterior uncertainty than this work (Table 3), which is not really discussed in the manuscript.

In my view, the conclusions and abstract should reflect these concerns with respect to these final emission estimates, which the authors themselves raise. I appreciate this study as a first step that covers a lot of ground, but, based on the presented analysis, I do not think that the Zurich emission estimatecan be trusted within the currently stated posterior uncertainties.

- 3. Given that the authors present an impressive amount of work, I would not suggest additional inversions to be included. But I would like to see a more start-to-end discussion of what the major limitations of the current framework are, and, most importantly, how (your) future work can address these. A non-exhaustive list:
  - a. The influence of the biosphere, which is discussed as being too low. As mentioned, there are some simple sensitivity tests you can do by scaling up the prior biosphere fluxes. Additionally, there is an urban version of VPRM that includes, e.g., surface porosity and the urban heat island effect, likely to result in higher NEE as well (e.g., Hardiman et al., 2017).
  - b. Are there any potential steps forward you can take in separating biosphere and anthropogenic fluxes?
  - c. The problematic winter observations that are not well-simulated could be filtered out. Additionally, one could consider including filtered nighttime observations (Monteiro et al., 2024), although I understand that's quite a new development. E.g., in Lian et al., 2023, filtered morning observations are included.
  - d. Given the limitations of EnKF posterior uncertainties, sensitivity tests can help quantify the uncertainties better.
  - e. Any key lessons on comparing emission estimates for different cities? To my knowledge this is the first urban inversion that includes more than one city.

**Other points:**

- The introduction doesn't explain previous literature in enough detail. For example, what specifically does previous work say about separating anthropogenic emissions from biosphere fluxes, beyond that it is difficult? Given that Paris has

quite an extensive history of high-quality urban inversion work, I would also expect some more specific discussion beyond that your posterior totals agree. Generally, I would expand the introduction with specific, relevant findings from previous urban studies. It also meanders into too much detail in L44-48, which is more fitting for the methods (see next point).

- The manuscript is very long. At times this is necessary, but not everywhere. I think that some adjustments can strengthen the presentation of the most important results. Some suggestions:
  - Are all 15 figures required to be in the main text? Fig. 10, 13 could be supplementary in my view. Possibly others at discretion of the authors.
  - Much of the discussion of the results outside Zurich and Paris (i.e., the European domain) could be supplementary, with a few sentences addressing this in the main text.
  - L254-268: Given that off-diagonal errors for R are not included, they do not need to be explained in this much detail.
  - L281-286: This can be 1 or 2 sentences.
  - L311-315: This is just aggregation of uncertainties, I don't think it needs to be explained.
  - o Conclusions are very verbose.

These are non-exhaustive, I think it would help the manuscript's readability if it were more concise.

- It is common practice to include only afternoon observations in CO2 inversions. However, since the overpass time of air over a city is short, afternoon observations only inform on afternoon fluxes, and perhaps late morning. I would like to see a brief discussion of the potential impact on flux estimation, given that it can depend on a city's size and diurnal cycle in fluxes. E.g., do you completely miss the morning rush hour? How high are nighttime fluxes (anthropogenic and biosphere), and what is their contribution to the CO2 budget? What are the consequences of scaling the non-afternoon fluxes along with the afternoon fluxes, when only the latter are constrained by observations?

**Minor points:**

- L20-22: While it is true that there are not that many urban inversions, there are a lot more than these 4, so I suggest to add: "for example".
- L22-23: "cities are major contributors" Please add reference and possibly an estimated number (e.g., 44% from Seto et al., 2014).
- L44-48: This is too detailed for the introduction, it is more fitting for methods.

- Why is the prior error for anthropogenic emissions higher (100%) than for biosphere fluxes (50%)? It is repeatedly mentioned how uncertain biosphere fluxes are.
- Fig. 2: Maybe mark resolution with e.g., 1 km x 1 km, so it is clear that it is a resolution. Also, I am not sure what the dashed lines in the inner figure mark without x/y labels, so I suggest removing them. Given that no coordinates are given, a scale in the inner zooms would help: it's easier to interpret than the area number, and useful for comparing to the spatial correlation scale.
- What is the temporal resolution of scaling factors in the state? Is it one per week?
- Why are the errors in R smoothed over 5 weeks? How does that affect the impact of anomalously high model-data differences in winter that are currently affecting the inversion result quite a bit?
- L266: How much is the impact? I would expect that correlations between sites reduce the information in the system by a lot, so I am surprised by this result.
- L339: Mention that it's Pearson r at first mention of the correlation, no need to mention the python function.
- L380-381: How so? Is the next paragraph describing this unrealistic behavior? Could be more explicit.
- L385-386: I do not follow this logic. If anthropogenic emissions and biosphere were interchangeable, then the inversion would stick to the prior, since that gives a lower cost function. Therefore, the fact that it is adjusting the two in opposite directions means that they are to some extent separable. The question is: is this real or does it have to do with e.g., overfitting?
- L424: Suggest Paris instead of French capital.
- Fig. 12c: These local reductions are huge, scaling down certain grid cells by >50%, and that is on the annual mean where the text mentions it mostly comes from winter. Is there any reason to believe these reductions are realistic or does it only have to do with transport errors? Does it fully disappear in summer?
- Fig. 14: Expand description. E.g., RMSE/bias between simulated and actual observations.
- I would like to see a supplementary figure with spatial distributions of anthropogenic emissions, GPP and RE and their diurnal cycle in the supplements, given that this is the information that the system allows you to separate the three (L39-40). Partly this information is already in existing figures, but not everything (e.g., diurnal cycles, or GPP and RE split) and not in a way that makes it easy to compare.

**References**

1. Seto, K.C., Dhakal, S., Bigio, A., Blanco, H., Carlo Delgado, G., Dewar, D., Huang, L., Inaba, A., Kansal, A., Lwasa, S. and McMahon, J., 2014. Human settlements, infrastructure, and spatial planning.

- 2. Hardiman, B.S., Wang, J.A., Hutyra, L.R., Gately, C.K., Getson, J.M. and Friedl, M.A., 2017. Accounting for urban biogenic fluxes in regional carbon budgets. Science of the Total Environment, 592, pp.366-372.
- 3. Sargent, M., Barrera, Y., Nehrkorn, T., Hutyra, L.R., Gately, C.K., Jones, T., McKain, K., Sweeney, C., Hegarty, J., Hardiman, B. and Wang, J.A., 2018. Anthropogenic and biogenic CO2 fluxes in the Boston urban region. Proceedings of the National Academy of Sciences, 115(29), pp.7491-7496.
- 4. Lian, J., Lauvaux, T., Utard, H., Bréon, F.M., Broquet, G., Ramonet, M., Laurent, O., Albarus, I., Chariot, M., Kotthaus, S. and Haeffelin, M., 2023. Can we use atmospheric CO2 measurements to verify emission trends reported by cities? Lessons from a 6-year atmospheric inversion over Paris. Atmospheric Chemistry and Physics, 23(15), pp.8823-8835.
- 5. Monteiro, V.C., Turnbull, J.C., Miles, N.L., Davis, K.J., Barkley, Z. and Deng, A., 2024. Assimilating morning, evening, and nighttime greenhouse gas observations in atmospheric inversions. Journal of Geophysical Research: Atmospheres, 129(17), p.e2024JD040998.

---

## Author Comment (AC1)

**Reply to reviewer #1**

We would like to thank the reviewer for the critical reading and valuable suggestions. In the following, we address the points one by one presenting our replies in dark blue and changes to the manuscript in dark red.

The manuscript titled "Estimation of CO2 fluxes in the cities of Zurich and Paris using the ICON-ART CTDAS inverse modelling framework" is a comprehensive paper presenting a sophisticated inverse modelling study using the ICON-ART CTDAS framework to estimate urban CO2 fluxes in Zurich and Paris. The topic is highly relevant for the ICOS Cities project and for ongoing efforts to develop observation-based methods for tracking urban greenhouse gas emissions. The manuscript is clearly structured, methodologically sound, and rich in technical detail. It contributes significantly to advancing urban-scale inverse modelling capabilities in Europe.

**Minor comments:**

**Figures8-11:**

It would be good to add shaded bands for one-sigma uncertainty to help visualize the posterior improvement. Label axes clearly with units.

We agree that visualizing uncertainty with shaded bands improves clarity. Shaded bands showing one-sigma posterior uncertainties have been added to Figures 8–9 to illustrate the reduction in uncertainty after the inversion. We decided not to add uncertainty values to Figure 10, as it is already quite busy with eight lines indicating background updates for each of the eight different wind directions. For Figure 11, we added a note to the caption in order to clarify how the uncertainties were calculated: 'Uncertainties were propagated from weekly flux uncertainties as described in the Data and Methods section'. We have added labels with units where they were missing.

**Sect.3.1.2:**

The text could benefit from a concise comparison of model—observation statistics before and after inversion in a summary table.

We added a concise comparison of model—observation statistics before and after inversion in a summary table (new Table 3). This includes mean bias, RMSE, and Pearson correlation coefficients for both Zurich and Paris, highlighting the improvements due to the inversion.

**Technical comments**

The manuscript is generally well written. A few sentences in Sect 4 are lengthy and could be split for clarity (especially around lines 440–460).

We have split several long sentences in Section 4 (lines 440–460) to improve readability.

| We hope that these revisions address the reviewer's suggestions and overall improve the clarity of the manuscript. |
|--------------------------------------------------------------------------------------------------------------------|
|                                                                                                                    |
|                                                                                                                    |
|                                                                                                                    |
|                                                                                                                    |
|                                                                                                                    |
|                                                                                                                    |
|                                                                                                                    |
|                                                                                                                    |
|                                                                                                                    |

---

## Author Comment (AC2)

**Reply to Reviewer #2**

We would like to thank the reviewer for the careful reading of our manuscript and a lot of constructive comments. In the following, we address all points raised by the reviewer, with our replies shown in dark blue and the corresponding changes to the manuscript shown in dark red.

**Review of Ponomarev et al. (2025)**

This study presents an investigation of urban CO2 fluxes in Zurich and Paris through inverse modeling. It is interesting that the same modeling framework is applied to two different cities with different sizes, emission distributions and set-ups of their measurement network. However, I have some concerns regarding the set-up of the inversions, which are mostly mentioned in the paper, but not addressed in any type of sensitivity tests. I appreciate that the study covers a lot of ground, making it hard to dig deep everywhere. However, in the absence of sensitivity tests, the final emission estimates need to be presented more carefully. They should be accompanied by a more comprehensive discussion of the limitations and uncertainties, with suggestions on how these can be addressed in future work.

**My primary concerns:**

**1)** I think the authors undersell how complex their inversion set-up is compared to previous urban work. The system co-optimizes anthropogenic emissions and GPP and RE and boundary conditions. Previous work has shown that co-optimizing anthropogenic emissions and NEE with only CO2 observations is already difficult, if not situationally impossible (e.g., Sargent et al., 2018). This is one of the key difficulties in urban inversions. Yet, in most of the manuscript, the authors consider that all these components can be separately estimated, which is something that, to my knowledge, has never been shown before. It would therefore be important that the authors address how good their estimates are. The authors do address that the biosphere fluxes are likely too low in the prior, which makes co-optimization easier. Then, however, I think it would be an easy sensitivity test to run the inversion with biosphere prior x2 or x5. Additionally, given the inversion method, the authors can investigate the posterior covariances to see if estimation of GPP, RE and anthropogenic emissions are correlated. The current manuscript does not include analyses of this novel aspect of the study. I would really like to see some analysis of posterior covariances between flux categories, which is possible without additional inversions. As a specific example, I find it hard to believe that separate estimation of RE and GPP is possible, and that the interpretation of these two separately is meaningful (L490-493), especially with only afternoon observations. My hypothesis would be that the posterior covariances will indicate that the solutions for RE and GPP are correlated, and if this is not the case I would be very interested in the authors' interpretation of this result.

We agree that separating GPP and RE is challenging, especially when assimilating only afternoon observations. Our intent was not to claim full separability, but rather to add flexibility to the inversion system. The main motivation for estimating RE and GPP separately came from previous experiences with an inversion in another project (unpublished), where we applied the same scaling factors to GPP and RE (and were thus only optimizing NEE). We noticed in this inversion that it was unable to correct for the unrealistically large annual mean uptake of $CO_2$ by vegetation simulated by (that version of) VPRM. Another motivation was the findings of the

studies by Tolk et al. (2011) and White et al. (2019), which compared the results of separately optimizing GPP and RE versus optimizing only NEE in synthetic model experiments with a known truth. They concluded that the separate estimation of the two components performed better. When optimizing NEE, we need to scale GPP and RE with the same scaling factors. This preserves potentially wrong relative magnitudes. Another disadvantage of optimizing only NEE is that NEE can be small even when the individual components are not small but exactly compensating each other. In these situations it is essentially impossible to change NEE with a scaling approach.

We will emphasize the novelty and challenges of our approach, by adding the following note at the end of 2.3.2 State vector section:
'... Treating GPP and RE as separate components in the state vector is a new approach to urban inverse modeling. Separating these two components is generally difficult, and it is particularly challenging when only daytime observations are assimilated. Nevertheless, the study by Tolk et al. (2011) suggested that optimizing scaling factors for GPP and RE separately performs better than optimizing scaling factors for NEE only even when daytime observations are used. Their conclusion was based on evaluating six different regional $CO_2$ flux inversion approaches in synthetic model experiments. Similar conclusions were drawn also by White et al. (2019). By estimating GPP and RE separately, we provide the inversion more flexibility to adjust the biogenic fluxes. Optimizing only NEE would have required applying the same scaling factors to GPP and RE, which would preserve any prior errors in the relative magnitudes of the two components. We will present separate results for GPP and RE, but we will also discuss the degree to which they can be separated by analyzing their correlations in the posterior covariances.'

In the discussions sections, we added:
'... Posterior correlations between the two components (0.61 in Zurich, 0.58 in Paris) indicate partial coupling, reflecting that the inversion provides some flexibility to adjust GPP and RE separately. However, these correlations also show that GPP and RE are not fully independent in the inversion. Part of the observational constraint affects both components in a similar way. In this sense, separate optimization mainly prevents errors in the prior relative magnitudes from being preserved, but it does not imply that the system can always distinguish their individual variations. The strength of this separability varies significantly over the year and depends on the available observational information and atmospheric conditions. These limitations are not captured by the analytical posterior uncertainties and will require performing sensitivity tests.'

and in the Conclusions section we added:'... In addition, the posterior correlations between GPP and RE scaling factors (0.61 in Zurich, 0.58 in Paris) suggest that these components can be at least partially disentangled with the current inversion framework. Future work should include nighttime $CO_2$ observations during certain nights where atmospheric boundary layers are not very low to improve the separability of the components. Furthermore, targeted sensitivity tests with perturbed biospheric priors and with optimizing only NEE versus separately optimizing GPP and RE should be performed to further evaluate the robustness of the biogenic flux estimates.'.

**2)** With the lack of sensitivity tests, the authors rely on the posterior error statistics. While these have value, they are very optimistic as absolute uncertainties. I think the authors should be much more explicit that these statistics provide a limited view of the real uncertainties, which generally require sensitivity tests to be quantified. Reading the abstract would give the reader the (in my view false) impression that there are tiny posterior uncertainties and the system works incredibly well. In reality, many major uncertainties are not included in the current uncertainty estimate. In fact, my interpretation is that for Zurich the authors find that in winter the estimate is biased due to transport errors, and in summer the estimate is biased due to interference from the biosphere. Lian et al. (2023) also report a 3x higher posterior uncertainty than this work (Table 3), which is not really discussed in the manuscript. In my view, the conclusions and abstract should reflect these concerns with respect to these final emission estimates, which the authors themselves raise. I appreciate this study as a first step that covers a lot of ground, but, based on the presented analysis, I do not think that the Zurich emission estimate can be trusted within the currently stated posterior uncertainties.

The reviewer correctly pointed that the reported uncertainties could be misleading as they only represent the analytical posterior uncertainties captured within the ensemble Kalman inversion. We are aware of the fact that structural and parametric uncertainties, which typically lead to systematic errors, can add substantially to the overall uncertainty. We discussed this in previous publications (e.g. Brunner et al., 2017; Henne et al., 2016). In the study of Henne et al. (2016), for example, we performed an ensemble of inversions with different settings to better capture these uncertainties. Unfortunately, the high computational cost of our inversion currently prevents us from running an ensemble of inversions, or it would require reducing model resolution considerably.

For a more complete uncertainty assessment, dedicated sensitivity tests would indeed be helpful, as suggested by the reviewer. Such tests could include scaling prior biospheric fluxes, perturbing transport model settings, or modifying observation error covariance matrices. We added a few notes about these limitations after Table 4 in the Discussions:
'... We note, however, that the posterior uncertainties in our work only reflect the analytical uncertainty of a Bayesian inversion, which assumes that all errors can be represented by Gaussian distributions with zero mean, i.e. all prior assumptions are unbiased. They do not include other sources of uncertainty such as transport model errors or representation errors introduced by comparing simulations with a bulk land surface scheme, which represents cities only through enhanced surface roughness but without a vertical canopy structure, with observations above rooftops. As a consequence, our reported posterior uncertainties are too low, but without an extensive analysis of potential additional errors, we do not know by how much. Our uncertainties reported for Paris are lower than in previous studies (e.g., Lian et al., 2023}, which also only accounted for analytical uncertainties but used different inversion approaches, prior uncertainties, and different spatial resolutions.'
Furthermore, we added the following sentence to the Conclusions:

'... Here we emphasize that the reported posterior uncertainties only reflect the inversion statistics and do not capture other sources of error. They should therefore be considered as lower bounds of the total uncertainty.'

**3)** Given that the authors present an impressive amount of work, I would not suggest additional inversions to be included. But I would like to see a more start-to-end discussion of what the major limitations of the current framework are, and, most importantly, how (your) future work can address these. A non-exhaustive list:

a. The influence of the biosphere, which is discussed as being too low. As mentioned, there are some simple sensitivity tests you can do by scaling up the prior biosphere fluxes. Additionally, there is an urban version of VPRM that includes, e.g., surface porosity and the urban heat island effect, likely to result in higher NEE as well (e.g., Hardiman et al., 2017).
b. Are there any potential steps forward you can take in separating biosphere and anthropogenic fluxes?
c. The problematic winter observations that are not well-simulated could be filtered out. Additionally, one could consider including filtered nighttime observations (Monteiro et al., 2024), although I understand that's quite a new development. E.g., in Lian et al., 2023, filtered morning observations are included.
d. Given the limitations of EnKF posterior uncertainties, sensitivity tests can help quantify the uncertainties better.
e. Any key lessons on comparing emission estimates for different cities? To my knowledge this is the first urban inversion that includes more than one city.

These are very good suggestions.

a.
We agree that more studies are required to improve the biogenic flux estimates. As we describe in the paper, in Paris, the CORINE land-use dataset poorly represents street trees and small green spaces. In Zurich we used a self-developed land-use dataset with much better representation of the urban vegetation. Nevertheless, prior NEE seems biased in the central parts of the city. We have already evaluated VPRM in the version used in this paper against measurements in Zurich (Stagakis et al. 2025) and our colleagues at the Technical University of Munich are working on improved formulations and more extensive evaluations in the cities of Munich and Zurich (a manuscript has been submitted). In their study, different variants of VPRM were tested including the urban version of Hardiman et al. (2017). The results did not indicate that urban VPRM performed better than the version used here. It should be noted that still very limited observations are available for evaluating biospheric models in cities.
We added the following sentence to Section 2.2.1 introducing VPRM after line 159:

'... The same parameters were also used in the study of Stagakis et al. (2025), which compared four different biospheric flux models applied over the city of Zurich with each other and against observations of respiration fluxes, leaf area density and sap flow in urban parks. This limited and partially indirect evaluation (e.g. sap flow used as a proxy of GPP) showed VPRM to

perform equally well as other, more complex biospheric models. A more extensive evaluation of different variants of VPRM including urban VPRM (Hardiman et al., 2017) against measurements in Zurich and Munich is currently ongoing. It is clear that further developments are needed to improve biospheric flux models for urban areas, since they have mostly been developed for, and tuned to, natural environments.'

Furthermore, we added the following sentence to the conclusions:
'... Further studies should explore higher-resolution urban vegetation datasets or urban-adapted models, such as urban VPRM (Hardiman et al., 2025), to better constrain the biogenic contribution. Unfortunately, as pointed out by Stagagiks et al. (2025), in-situ observations of urban vegetation are still too limited to thoroughly evaluate biospheric flux models in cities.'

b.
Several steps could be taken to improve the separation of biospheric and anthropogenic fluxes, such as incorporating additional tracers (e.g., radiocarbon or co-emitted species) and better constraining prior biogenic fluxes using vegetation models adapted to urban environments. Measurements of co-emitted species and radiocarbon were performed at the Hardau flux tower, but only over a few months and only in an experimental mode to explore their potential (see, Kunz et al., 2025). Including additional tracers comes with substantial challenges, which was beyond the scope of this study. We added the following sentence to the discussion section:
'... A better separation between anthropogenic and biospheric fluxes might be achieved through assimilation of additional tracers such as $NO_x$ or CO co-emitted with $CO_2$ or radiocarbon. Radiocarbon and co-emitted species have been measured as part of the ICOS Cities project at the Hardau tower but only for a few months and only using the Eddy covariance technique, which doesn't require very precise absolute calibration. Even if measurements would have been available for longer periods, the co-assimilation of $CO_2$ with other species comes with its own challenges, either because the species are chemically reactive (like $NO_x$), because emission ratios between these species and $CO_2$ are uncertain, or because the measurements are very challenging (as in the case of radiocarbon).'

c.
Wintertime or otherwise problematic observations could indeed be filtered to reduce the impact of transport biases. However, in the case of Zurich, several prolonged periods of low wind speeds occurred, which would have required removing all observations for several weeks, creating gaps in the inversion results. Instead, we opted to retain these observations but to increase the associated model uncertainties. Incorporating filtered nighttime or morning observations, as proposed by Monteiro et al. (2024) and Lian et al. (2023), appears very promising. At the same time, we are convinced that improving the representation of the urban meteorology, particularly for low wind speed periods in complex terrain and near rooftops, should have the highest priority. One option would be to set up additional measurement infrastructure such as doppler lidars capturing vertical wind profiles and to assimilate these observations in a meteorological data assimilation framework.

d.

As discussed above, we agree with the reviewer that the presented EnKF posterior uncertainties underestimate total uncertainty. Sensitivity tests, such as scaling biospheric priors, modifying transport configurations, or perturbing observation error assumptions, are required to better quantify uncertainties.

e.
There are several lessons we learned from these inversion experiments for two cities. The description of these lessons  is currently scattered throughout the paper, so we decided to summarize them in one place in the conclusions. For this, we rewrote one of the Conclusion paragraphs:
'...The differences in posterior emissions between Zurich and Paris highlight several important lessons for urban inversions. First, inversion results are strongly city-specific: network layout, terrain complexity, and city size lead to distinct flux corrections, background updates, and model–data mismatches. Second, observational network design critically affects inversion sensitivity. Zurich's dense rooftop network, combined with complex terrain, amplifies transport-related biases, particularly during stagnant winter conditions with low wind speeds and shallow mixing layers, which can inflate simulated concentrations and misattribute accumulated $CO_2$ to local sources. In contrast, Paris tower network and flatter terrain produce smoother regional constraints. These conditions make the inversion less sensitive to both transport errors and biogenic fluxes. Finally,  background adjustments also differ between the cities. Paris, located closer to the Atlantic Ocean, is exposed to air masses less affected by European anthropogenic emissions. In contrast, Zurich lies further inland and is more influenced by continental sources, resulting in a more complex signal and larger corrections.'

**Other points:**

- The introduction doesn't explain previous literature in enough detail. For example, what specifically does previous work say about separating anthropogenic emissions from biosphere fluxes, beyond that it is difficult? Given that Paris has quite an extensive history of high-quality urban inversion work, I would also expect some more specific discussion beyond that your posterior totals agree. Generally, I would expand the introduction with specific, relevant findings from previous urban studies. It also meanders into too much detail in L44-48, which is more fitting for the methods (see next point).
A more detailed description of previous studies was added to describe the challenge of separating biogenic and anthropogenic fluxes:
'... Different attempts have been made in previous studies to separate biospheric fluxes from anthropogenic emissions. Additional challenges arise at the urban scale. For example, Lian et al. (2023) showed for Paris that coarse land-use vegetation data can underestimate urban biosphere activity considerably. They also mention the lack of eddy covariance measurements over urban vegetation to validate prior biospheric flux estimates. Sargent et al. (2018) demonstrated for Boston that the urban biosphere can take up more than half of the anthropogenic $CO_2$ signal during summer afternoons, demonstrating the importance of constraining biospheric fluxes alongside anthropogenic emissions. Similarly, synthetic inversions

over Salt Lake City and Indianapolis emphasized that prior assumptions on biospheric fluxes strongly affected posterior emissions and that dense, strategically placed measurement networks are crucial to disentangle overlapping signals (Kunik et al., 2019; Lauvaux et al., 2016).'.

We removed some of the sentences from L44-48 and moved them to the Section 2.2.1 Model Setup:
'... In our case, we could benefit from very detailed city inventories provided by the authorities of Zurich and Paris. However, temporal profiles were not always available, necessitating the use of generic time profiles in some cases. '

We have expanded the discussion on the comparison with previous studies. However, we note that a direct comparison of trends or daily fluxes is limited due to differences in network, model setup, and simulation period. Similar to Lian et al., 2023, dormant-period fluxes are better constrained, as biogenic fluxes in Paris retain high uncertainty even after the inversion:

"... Consistent with Lian et al. (2023), the inversion provides more reliable constraints during the dormant period, when biogenic fluxes are minimal. Even though our inversions did not assimilate morning observations as in Lian et al. (2023), the city-scale totals remain in good agreement with both the prior inventory and previous studies, highlighting the robustness of our anthropogenic emission estimates."

- The manuscript is very long. At times this is necessary, but not everywhere. I think that some adjustments can strengthen the presentation of the most important results. Some suggestions:
    o Are all 15 figures required to be in the main text? Fig. 10, 13 could be supplementary in my view. Possibly others at discretion of the authors.
    o Much of the discussion of the results outside Zurich and Paris (i.e., the European domain) could be supplementary, with a few sentences addressing this in the main text.
    o L254-268: Given that off-diagonal errors for R are not included, they do not need to be explained in this much detail.
    o L281-286: This can be 1 or 2 sentences.
    o L311-315: This is just aggregation of uncertainties, I don't think it needs to be explained.
    o Conclusions are very verbose. These are non-exhaustive, I think it would help the manuscript's readability if it were more concise.

We went through the manuscript and agree that certain parts can be streamlined. Below is the summary of the main changes:
•        We agree that the Figure 13 could be moved to the Supplement as suggested. However, we prefer to keep Figure 10 in the main text, as it presents the optimized background scaling

factors, which are an essential component of our inversion approach. Figure 10 illustrates one of the major inversion results and highlights key differences in background variability between Zurich and Paris.

•       The results for the EU simulation provide a justification for our approach of replacing full European-scale inversions with a computationally inexpensive background scaling. This section illustrates the level of background uncertainty and errors arising from EU-scale emission uncertainties. As the section is already concise, we prefer to retain it in the main text.

•       Two paragraphs about the R matrix were shortened to one: '... Uncertainties in the differences between modeled and observed mole fractions are described by the observation error covariance matrix R. It accounts for measurement, transport, and representation errors. Representation errors occur when a model with limited spatial resolution cannot resolve measurements at a single point. In most cases, this matrix is considered to be diagonal, implying statistical independence of model–observation differences. This assumption may be less valid for dense urban networks where stations lie within a few kilometers. While more complex formulations including off-diagonal terms have been proposed (e.g., Ghosh et al., 2021), short sensitivity experiments with such configurations in our system showed only minor effects on flux estimates. Therefore, due to the high computational cost of full EnSRF inversions, we used a diagonal R matrix (see Eq. 5) and assimilated observations serially.'

•       We shortened the introduction part of the section on the P matrix: '… The prior error covariance matrix P describes uncertainties related to prior flux and background scaling factors, as well as their correlations. These correlations effectively reduce the degrees of freedom, which is necessary to avoid overfitting. The observation networks often lack the density to independently constrain all state vector elements. The correlation length defines the spatial scale of independently resolvable structures and how uncertainties aggregate across the domain.'

•       We also removed a repeating point about transport errors in Zurich in the Conclusions section: '… Forward simulations were evaluated against $CO_2$ observations in all three domains. Simulated mole fractions for the European domain showed high (monthly mean) correlations with observations from the ICOS network, indicating that the model captures effectively day-to-day variability. Biases and RMSEs were of the order of a few ppm. Additionally, the model reproduced daily variations in wind speed and temperature well (see Supplementary Information). ~~However, comparisons revealed a strong sensitivity to wind speed, with mole fraction errors increasing substantially during calm periods, limiting the inversion performance in Zurich with its complex terrain. In contrast, over the domain of Paris, which is characterized by flatter terrain and a tower-based measurement network, instances of low wind speeds were less frequent and model observation mismatches correspondingly smaller.~~' We also rewrote a few other parts of the conclusions to make them less verbose.

- It is common practice to include only afternoon observations in CO2 inversions. However, since the overpass time of air over a city is short, afternoon observations only inform on afternoon fluxes, and perhaps late morning. I would like to see a brief discussion of the potential impact on flux estimation, given that it can depend on a city's size and diurnal cycle in fluxes. E.g., do you completely miss the morning rush hour? How high are nighttime fluxes (anthropogenic and biosphere), and what is their contribution to the CO2 budget? What are the

consequences of scaling the non-afternoon fluxes along with the afternoon fluxes, when only the latter are constrained by observations?

We agree that adding a short discussion of this common but limiting approach to the Discussion section is appropriate. We have therefore added the following: '... Finally, our inversions used only afternoon $CO_2$ observations, which primarily constrain daytime fluxes. Assimilating only observations when the boundary layer is fully developed is conventional practice as it minimizes errors introduced by a mismatch between simulated and real vertical mixing, which are largest when the boundary layer is low. This approach is further justified by the fact that nighttime and early-morning fluxes are lower than daytime although not negligible: in Zurich they account for roughly 30 % of the daytime flux, and in Paris for about 30-35 % (see Supplementary Fig. 10). At the same time, it complicates separation of RE and GPP, as during the night only one of the biospheric flux components is active.' In addition, we added a new supplementary figure showing the diurnal cycle of anthropogenic and biogenic fluxes to illustrate that nighttime fluxes are indeed low, while the morning rush hour falls outside the window of observations used for assimilation.

**Minor points**:

- L20-22: While it is true that there are not that many urban inversions, there are a lot more than these 4, so I suggest to add: "for example".

Added "for example" to the sentence.

- L22-23: "cities are major contributors" Please add reference and possibly an estimated number (e.g., 44% from Seto et al., 2014).
Added the suggested reference and estimated number.

- L44-48: This is too detailed for the introduction, it is more fitting for methods.

As mentioned earlier, we moved some of the sentences to the Section 2.2.1 Model Setup.

- Why is the prior error for anthropogenic emissions higher (100%) than for biosphere fluxes (50%)? It is repeatedly mentioned how uncertain biosphere fluxes are.

We used a 50% variance weight for each of the biospheric components (GPP and RE) in the prior covariance. Because the GPP and RE variance terms enter the prior covariance additively (and are treated as independent), the prior variance of NEE is approximately 0.5+0.5=1.0 (in the same variance-normalized units). In other words, the prior uncertainty on NEE is ~100% in our formulation.

- Fig. 2: Maybe mark resolution with e.g., 1 km x 1 km, so it is clear that it is a resolution. Also, I am not sure what the dashed lines in the inner figure mark without x/y labels, so I suggest removing them. Given that no coordinates are given, a scale in the inner zooms would help: it's

easier to interpret than the area number, and useful for comparing to the spatial correlation scale.

We have updated Fig. 2 to mark the resolution, removed the dashed grid lines from the inner domains, and added scale bars.

- What is the temporal resolution of scaling factors in the state? Is it one per week?

Yes, fluxes and background concentrations are adjusted weekly within the inversion framework.

- Why are the errors in R smoothed over 5 weeks? How does that affect the impact of anomalously high model-data differences in winter that are currently affecting the inversion result quite a bit?

Initially, we used monthly mean bias-corrected RMSE values, but large model–observation mismatches in Zurich during winter led us switch to weekly values. Weekly RMSE can fluctuate, so we applied smoothing over 5 weeks (results with 3 and 5 weeks were similar) to reduce abrupt changes in the model–data mismatch. This slightly reduced the strong flux adjustments in Zurich during that period, but ideally the simulations should be improved to better capture calm weather conditions.

- L266: How much is the impact? I would expect that correlations between sites reduce the information in the system by a lot, so I am surprised by this result.

The tests were limited to short assimilation windows and a few offline re-optimizations without rerunning the full ICON simulations. The inclusion of off-diagonal terms in R slightly altered the weekly flux adjustments, but the impact on annual mean fluxes for Zurich was below ~10%, with short-term differences occasionally reaching 15%. This modest sensitivity reflects the redundancy in the Zurich network, where several sites observe similar air masses.

- L339: Mention that it's Pearson r at first mention of the correlation, no need to mention the python function.

As suggested, the reference to the Python function was removed.

- L380-381: How so? Is the next paragraph describing this unrealistic behavior? Could be more explicit.

We have clarified the sentence to indicate what aspect of the posterior fluxes appears unrealistic: "…as the posterior fluxes showed excessively strong reductions in late December and early January."

- L385-386: I do not follow this logic. If anthropogenic emissions and biosphere were interchangeable, then the inversion would stick to the prior, since that gives a lower cost

function. Therefore, the fact that it is adjusting the two in opposite directions means that they are to some extent separable. The question is: is this real or does it have to do with e.g., overfitting?

We thank the reviewer for pointing this out. The opposite adjustments of anthropogenic and biospheric fluxes in summer indeed indicate that the inversion can partially separate the two sources. However, the separability is limited due to the relatively small magnitude of biospheric fluxes compared to anthropogenic emissions in Zurich and the use of afternoon-only observations. We have clarified this in the revised manuscript:'... The increase in posterior uptake in June was partially offset by higher anthropogenic emissions, indicating partial separability between anthropogenic and biospheric fluxes.'

- L424: Suggest Paris instead of French capital.

Changed "French capital" to "Paris" as suggested.

- Fig. 12c: These local reductions are huge, scaling down certain grid cells by >50%, and that is on the annual mean where the text mentions it mostly comes from winter. Is there any reason to believe these reductions are realistic or does it only have to do with transport errors? Does it fully disappear in summer?

These large local reductions mainly occur during winter and are largely driven by transport model uncertainties during low wind speed situations, particularly in residential areas outside the city center. In summer, the reductions largely disappear, as fluxes are smaller and transport is better represented.

- Fig. 14: Expand description. E.g., RMSE/bias between simulated and actual observations.

As suggested, we clarified the caption to indicate that the figure shows bias and RMSE between simulated and observed $CO_2$ mole fractions: 'Bias and RMSE of \chem{CO_2} mole fractions between simulated and observed values as a function of wind speed in Paris and Zurich.'

- I would like to see a supplementary figure with spatial distributions of anthropogenic emissions, GPP and RE and their diurnal cycle in the supplements, given that this is the information that the system allows you to separate the three (L39-40). Partly this information is already in existing figures, but not everything (e.g., diurnal cycles, or GPP and RE split) and not in a way that makes it easy to compare.

We have added new supplementary figures to address this: Figures S10 and S11 now show the diurnal cycles of anthropogenic emissions, GPP, and RE for Paris and Zurich, as well as maps of flux adjustments (posterior minus prior) also for all three flux components.

**References**

Seto, K.C., Dhakal, S., Bigio, A., Blanco, H., Carlo Delgado, G., Dewar, D., Huang, L., Inaba, A., Kansal, A., Lwasa, S. and McMahon, J., 2014. Human settlements, infrastructure, and spatial planning.

Hardiman, B.S., Wang, J.A., Hutyra, L.R., Gately, C.K., Getson, J.M. and Friedl, M.A., 2017. Accounting for urban biogenic fluxes in regional carbon budgets. Science of the Total Environment, 592, pp.366-372.

Sargent, M., Barrera, Y., Nehrkorn, T., Hutyra, L.R., Gately, C.K., Jones, T., McKain, K., Sweeney, C., Hegarty, J., Hardiman, B. and Wang, J.A., 2018. Anthropogenic and biogenic CO2 fluxes in the Boston urban region. Proceedings of the National Academy of Sciences, 115(29), pp.7491-7496.

Lian, J., Lauvaux, T., Utard, H., Bréon, F.M., Broquet, G., Ramonet, M., Laurent, O., Albarus, I., Chariot, M., Kotthaus, S. and Haeffelin, M., 2023. Can we use atmospheric CO2 measurements to verify emission trends reported by cities? Lessons from a 6-year atmospheric inversion over Paris. Atmospheric Chemistry and Physics, 23(15), pp.8823-8835.

Monteiro, V.C., Turnbull, J.C., Miles, N.L., Davis, K.J., Barkley, Z. and Deng, A., 2024. Assimilating morning, evening, and nighttime greenhouse gas observations in atmospheric inversions. Journal of Geophysical Research: Atmospheres, 129(17), p.e2024JD040998

Tolk, L. F., Dolman, A. J., Meesters, A. G. C. A., and Peters, W.: A comparison of different inverse carbon flux estimation approaches for application on a regional domain, Atmos. Chem. Phys., 11, 10349–10365, https://doi.org/10.5194/acp-11-10349-2011, 2011.

White, E. D., Rigby, M., Lunt, M. F., Smallman, T. L., Comyn-Platt, E., Manning, A. J., Ganesan, A. L., O'Doherty, S., Stavert, A. R., Stanley, K., Williams, M., Levy, P., Ramonet, M., Forster, G. L., Manning, A. C., and Palmer, P. I.: Quantifying the UK's carbon dioxide flux: an atmospheric inverse modelling approach using a regional measurement network, Atmos. Chem. Phys., 19, 4345–4365, https://doi.org/10.5194/acp-19-4345-2019, 2019.

Brunner, D., Arnold, T., Henne, S., Manning, A., Thompson, R. L., Maione, M., O'Doherty, S., & Reimann, S., 2017. Comparison of four inverse modelling systems applied to the estimation of HFC-125, HFC-134a, and $SF_6$ emissions over Europe. Atmospheric Chemistry and Physics, 17(17), 10651-10674. https://doi.org/10.5194/acp-17-10651-2017

Henne, S., Brunner, D., Oney, B., Leuenberger, M., Eugster, W., Bamberger, I., Meinhardt, F., Steinbacher, M., & Emmenegger, L., 2016. Validation of the Swiss methane emission inventory by atmospheric observations and inverse modelling. Atmospheric Chemistry and Physics, 16(6), 3683-3710. https://doi.org/10.5194/acp-16-3683-2016

Stagakis, S., Brunner, D., Li, J., Backman, L., Karvonen, A., Constantin, L., Järvi, L., Havu, M., Chen, J., Emberger, S., and Kulmala, L.: Intercomparison of biogenic CO2 flux models in four urban parks in the city of Zurich, Biogeosciences, 22, 2133–2161, https://doi.org/10.5194/bg-22-2133-2025, 2025.

Kunz, A.-K., Borchardt, L., Christen, A., Della Coletta, J., Eritt, M., Gutiérrez, X., Hashemi, J., Hilland, R., Jordan, A., Kneißl, R., Legendre, V., Levin, I., Preunkert, S., Rubli, P., Stagakis, S., & Hammer, S. (2025). A relaxed eddy accumulation flask sampling system for 14C-based partitioning of fossil and non-fossil $CO_2$ fluxes. Atmospheric Measurement Techniques, 18(20), 5349-5373. https://doi.org/10.5194/amt-18-5349-2025

Lian, J., Lauvaux, T., Utard, H., Bréon, F.-M., Broquet, G., Ramonet, M., Laurent, O., Albarus, I., Chariot, M., Kotthaus, S., Haeffelin, M., Sanchez, O., Perrussel, O., Denier van der

Gon, H. A., Dellaert, S. N. C., and Ciais, P. (2023). Can we use atmospheric CO2 measurements to verify emission trends reported by cities? Lessons from a 6-year atmospheric inversion over Paris, Atmos. Chem. Phys., 23, 8823–8835, https://doi.org/10.5194/acp-23-8823-2023